# Hexokinase II dissociation alone cannot account for changes in heart mitochondrial function, morphology and sensitivity to permeability transition pore opening following ischemia

Gonçalo C. Pereira[1¤]*, Laura Lee[2], Nadiia Rawlings[2], Joke Ouwendijk[1,3], Joanne E. Parker[1], Tatyana N. Andrienko[1], Jeremy M. Henley[2], Andrew P. Halestrap[1]

1 School of Biochemistry and Bristol Heart Institute, University of Bristol, Bristol, United Kingdom, 2 Centre for Synaptic Plasticity, School of Biochemistry, University of Bristol, Bristol, United Kingdom, 3 School of Biochemistry and Wolfson Bioimaging Facility, University of Bristol, Bristol, United Kingdom

¤ Current address: MRC – Mitochondrial Biology Unit, University of Cambridge, Cambridge, United Kingdom
* g.pereira@mrc-mbu.cam.ac.uk

## Abstract

We previously demonstrated that hexokinase II (HK2) dissociation from mitochondria during cardiac ischemia correlates with cytochrome c (cyt-c) loss, oxidative stress and subsequent reperfusion injury. However, whether HK2 release is the primary signal mediating this ischemia-induced mitochondrial dysfunction was not established. To investigate this, we studied the effects of dissociating HK2 from isolated heart mitochondria. Mitochondria isolated from Langendorff-perfused rat hearts before and after 30 min global ischemia ± ischemic preconditioning (IPC) were subject to *in vitro* dissociation of HK2 by incubation with glucose-6-phosphate at pH 6.3. Prior HK2 dissociation from pre- or end-ischemic heart mitochondria had no effect on their cyt-c release, respiration (± ADP) or mitochondrial permeability transition pore (mPTP) opening. Inner mitochondrial membrane morphology was assessed indirectly by monitoring changes in light scattering (LS) and confirmed by transmission electron microscopy. Although no major ultrastructure differences were detected between pre- and end-ischemia mitochondria, the amplitude of changes in LS was reduced in the latter. This was prevented by IPC but not mimicked *in vitro* by HK2 dissociation. We also observed more Drp1, a mitochondrial fission protein, in end-ischemia mitochondria. IPC failed to prevent this increase but did decrease mitochondrial-associated dynamin 2. *In vitro* HK2 dissociation alone cannot replicate ischemia-induced effects on mitochondrial function implying that *in vivo* dissociation of HK2 modulates end-ischemia mitochondrial function indirectly perhaps involving interaction with mitochondrial fission proteins. The resulting changes in mitochondrial morphology and cristae structure would destabilize outer / inner membrane interactions, increase cyt-c release and enhance mPTP sensitivity to [$Ca^{2+}$].

**Data Availability Statement:** All relevant data are within the paper and its Supporting Information files.

**Funding:** This work was supported by Programme and Project Grants from the British Heart Foundation (RG/08/001/24717 to APH, PG/12/40/29634 to APH, and PG/14/60/3014 to JMH). There was no additional external funding received for this study. The funding agency and the University had no role in study design, data collection and analysis, decision to publish, or preparation of the manuscript.

**Competing interests:** The authors have declared that no competing interests exist.

**Abbreviations:** ANT, adenine nucleotide translocator; AUC, area under the curve; BKA, bongkrekic acid; CAT, carboxyatractyloside; CrK, creatine kinase; CsA, cyclosporine A; Drp1, dynamin-related protein 1; Dyn2, dynamin 2; G6P, glucose-6-phospate; GMS, 5 mM L-glutamate, 2 mM L-malate and 5 mM succinate; HK, hexokinase; I/R, ischemia-reperfusion; IMM, inner mitochondrial membrane; IPC, ischemic preconditioning; Mff, mitochondrial fission factor; mPTP, mitochondrial permeability transition pore; mt-HK, mitochondrial-bound hexokinase; OMM, outer mitochondrial membrane; OMMP, outer mitochondrial membrane permeabilization; PCr, phosphocreatine; TEM, transmission electron microscopy.

# Introduction

Reperfusion injury occurs upon restoration of blood flow after a prolonged period of ischemia. It has major clinical significance because reperfusion is essential to restore heart function yet may precipitate further irreversible injury, and thus limit the extent of myocardial salvage. A key player in ischemia-reperfusion (I/R) injury is the opening of a non-selective, high conductance channel in the inner mitochondrial membrane (IMM) known as the mitochondrial permeability transition pore (mPTP) [1] whose exact molecular identity remains uncertain [2–5]. It has been widely accepted that the major triggers for mPTP opening during reperfusion are elevated $[Ca^{2+}]_m$ and reactive oxygen species (ROS) [1]. It has been further proposed that ROS are generated early in reperfusion by reverse electron flow through Complex I of the respiratory chain during oxidation of succinate that accumulates in ischemia [6]. However, our own real-time surface fluorescent measurements of ROS during reperfusion do not support this hypothesis [7]. Indeed, ischemic preconditioning (IPC), in which hearts are exposed to several brief periods of ischemia and reperfusion prior the prolonged ischemic episode, is strongly cardioprotective but is not accompanied by a reduction in succinate accumulation [7, 8]. Furthermore, Cyclosporine A (CsA), a specific inhibitor of the mPTP that acts by binding to an established regulatory component of the pore (cyclophilin D), has failed to show reduction in mortality and adverse effects of I/R in clinical settings [9] despite its cardioprotection in many animal models. These observations raise the question whether simple inhibition of the mPTP is sufficient to prevent I/R injury and whether targeting events upstream of mPTP opening might be more effective.

In this respect, we have previously shown that although the mPTP remains closed during ischemia [10], events occurring during this period sensitize mitochondria to mPTP opening at reperfusion [11]. Thus, mitochondria isolated from ischemic hearts are much more sensitive to mPTP opening than those from control hearts. However, IPC or pharmacological preconditioning attenuates this sensitization [11, 12] and thus leads to less mPTP opening on reperfusion and consequently less injury. More recently [7] we showed that IPC did not attenuate the elevated $[Ca^{2+}]_m$ at the end of ischemia or ROS production in the first 90 s of reperfusion yet mPTP opening after 1 min and subsequent ROS production was attenuated. This confirms that mitochondria experience an effect during ischemia that determines mPTP opening and thus the extent of injury on reperfusion. In this regard, ischemia is associated with a decrease in the amount of hexokinase 2 (HK2) bound to mitochondria which is caused by accumulation of glucose-6-phosphate (G6P) and a drop in cytosolic pH that occurs as a result of enhanced glycolysis during this phase [13]. IPC attenuates these metabolic changes and prevents the dissociation of mitochondrial-bound HK2 (mt-HK2), suggesting this may be an important component of the IPC mechanism. In this context it should be noted that mitochondria-bound HK2 has also been implicated in the resistance of tumour cell mitochondria to mPTP opening [14–17]. Furthermore, extensive earlier studies from our laboratory [13, 18, 19] and those of others [20–23] have demonstrated a strong correlation between dissociation of mt-HK2 during ischemia and cytochrome c release and infarct size at reperfusion. In this regard, cytochrome c loss decreases superoxide scavenging and promotes ROS production [19] which are known inducers of the mPTP [1]. By contrast, IPC does not reduce the accumulation of succinate, a proposed source of ROS, during ischemia or the rate of its subsequent decline on reperfusion [7, 8] but does prevent HK2 loss from mitochondria during ischemia [13]. Yet, despite the numerous reports about hexokinase(s) on cell survival, either in the context of cardiovascular injury and/or cancer, none of these papers provide definite evidence of whether or how HK2 can modulate mPTP opening directly. Rather they have primarily investigated upstream events leading to mt-HK2 release, such as the potential roles of Akt and GSK3beta, reporting

mt-HK2 dissociation as the last event in a cascade that leads to mPTP opening. They do not provide evidence that HK2 directly regulates mPTP opening.

Previous data from our laboratory have offered some insights into the mechanism by which mt-HK2 might interact with the mPTP. We observed that rates of phosphocreatine (PCr) regeneration following prolonged ischemia are depressed [13], suggesting a disruption of the PCr-dependent energy shuttling system that involves interactions between the outer and inner mitochondrial membranes (OMM and IMM). Another hetero-protein complex found at points of juxtaposition between OMM and IMM comprises the adenine nucleotide translocase (ANT), the voltage-dependent anion channel (VDAC) and, importantly for the present study, HK [24]. Interestingly, we [25] and others [26] have reported that modulation of such contact sites in isolated mitochondria is associated with cytochrome c release and increased sensitivity to mPTP opening. These data suggest a plausible role for OMM and IMM interactions in linking extra-mitochondrial events that occur during ischemia to sensitization of mPTP opening during I/R.

The primary aim of this study was to establish whether the deleterious effects of ischemia on mitochondria *in vitro* can be mimicked purely by displacing mt-HK2. Using a brief incubation with G6P at pH 6.3 during mitochondrial isolation, we have dissociated HK2 from mitochondria isolated from pre-ischemic and end-ischemic control and IPC hearts and subsequently measured OMM permeabilization (OMMP), the sensitivity of the mPTP and changes in OMM and IMM morphology. Our rationale is that, if prevention of HK2 dissociation from mitochondria is directly responsible for the attenuation of changes to mitochondria function at the end of ischemia, then its removal from IPC end-ischemic mitochondria should change these parameters back to those of non-preconditioned end-ischemic mitochondria. However, our data reveal that HK2 dissociation *in vitro* does not replicate ischemia-induced effects on these parameters which leads us to conclude that release of mitochondrial-bound HK2 *in vivo* must affect mitochondrial morphology and function indirectly. We present preliminary evidence that this may be achieved by modulating the binding of the mitochondrial fission proteins Drp1 and mitochondrial-associated dynamin 2 (Dyn2).

## Material and methods

### Animals

Male Wistar rats, RccHan:WIST (225–275 g), were purchased from Harlan (Oxfordshire, UK), acclimatized for 5–7 days prior to the initiation of experiments and maintained in the local animal house facility (Biomedical Sciences Building, University of Bristol, Bristol, UK). Animals were group-housed in grilled top type IV cages with corncob grit bedding and environmental enrichment (irradiated tissue paper and cardboard for nest building and shelter) as means of improving animal welfare. Cages were maintained under controlled environmental requirements (22 ˚C, 45–65% humidity, 15–20 air changes/hour, 12 h artificial light/dark cycle, noise level < 55 dB); and, free access to rodent food (EURodent Diet 22%, 5LF5, Labdiet, St. Louis, MO, US) and water ad libitum. Animals were euthanized by concussion followed by cervical dislocation, and hearts were immediately extracted from the body and quickly immersed in ice-cold Krebs–Henseleit buffer.

Animal handling was performed in accordance with the European Convention for the Protection of Vertebrate Animals used for Experimental and Other Scientific Purposes (CETS no.123), the UK Animals (Scientific Procedures) Act 1986 amendment regulations 2012 and was approved by the appropriate University of Bristol ethics committee (UB/09/012). G.C.P. is credited by the European Federation for Laboratory Animal Research (FELASA) category C for animal experimentation (accreditation no. 020/08).

## Heart perfusion

Langendorff retrograde heart perfusion was performed as previously described [18] using constant flow (12 mL/min) with Krebs–Henseleit buffer (NaCl 118 mM, NaHCO$_3$ 25 mM, KCl 4.8 mM, KH$_2$PO$_4$ 1.2 mM, MgSO$_4$ 1.2 mM, glucose 11 mM and CaCl$_2$ 1.2 mM) gassed with 95% O$_2$ / 5% CO$_2$ at 37 ˚C. After cannulation, hearts were perfused for 35 min followed by 30 min global normothermic ischemia (index ischemia), induced by halting perfusion and immersing the heart in perfusion buffer at 37 ˚C. Ischemic pre-conditioning (IPC) was induced after 20 min perfusion by three cycles of 2 min ischemia plus 3 min reperfusion before entering index ischemia as described above. Hemodynamic monitoring was achieved by inserting a latex balloon connected to a pressure transducer into the left ventricle. Data acquisition and analysis were performed using PowerLab/LabChart (AdInstruments, Oxford, UK). A summary of the hemodynamic data is presented in the supplemental S1, S2 and S3 Tables.

For hearts used to assess the content of mitochondrial dynamic proteins (ie., Drp1, Mff and Dyn2) the standard perfusion protocol was changed as following: control perfusion (pre-ischemia), 45 min perfusion; end-ischemia, 15 min stabilisation followed by 30 min global normothermic ischemia; IPC, 15 min stabilisation followed by 3 cycles of 2 min bursts of ischemia separated by 3 min perfusion, then 30 min global normothermic ischemia.

## Assessment of infarct size

For assessment of infarct size hearts were reperfused for 120 min after ischemia before perfusing for 2 min at 10 mL/min with a 1% (w/v) TTC solution. Hearts were then detached from the cannula and incubated for an additional 5 min at 37 ˚C before being sliced perpendicular to the longitudinal axis into 6 slices. The slices were then fixed in 4% (w/v) formalin solution overnight at 4˚C. Both sides of each slice were imaged on a standard office scanner and acquisition settings were maintained for all analysed hearts. The surfaces of the necrotic and area at risk of each side for each slice were determined by planimetry using a color threshold in ImageJ (NIH, Bethesda, MD, USA). Because global ischemia was used, infarct size was expressed as a percentage of the total cross-sectional area of the heart. A summary of the data obtained is shown in the supplemental S3 Table.

## Mitochondria isolation

Hearts were removed from the cannula either before entering ischemia (pre-ischemia) or at the end of ischemia, and immediately immersed in ice-cold isolation buffer (sucrose 300 mM, EGTA 2 mM, and Tris-HCl 10 mM, pH 7.3 at 4 ˚C). Mitochondria were isolated through differential centrifugation as described previously [18], with slight modifications. All steps were performed at 4 ˚C. Briefly, hearts were finely chopped with a razor blade on a glass petri dish and incubated at 4 ˚C for 4.5 min with stirring in a small beaker with 6 mL of isolation buffer supplemented with 1.12 U/mL of protease (EC 3.4.21.62, Subtilisin A) from Bacillus licheniformis (aqueous propylene glycol solution from Sigma-Aldrich, Dorset, UK). The suspension was then filtered through a nylon mesh (100 μm) and washed with 7 volumes of isolation buffer before transfer into a glass Potter homogenizer. Homogenization was carried out in 25 mL of isolation buffer supplemented with 0.5% defatted BSA for about 2 min using a motorized Teflon pestle. The homogenate was centrifuged at 7,500 g for 7 min, and the resulting pellet was re-suspended in 25 mL of isolation buffer with BSA and subjected to additional homogenization as described earlier. The suspension was then centrifuged at 700 g for 10 min and the resultant supernatant saved while the pellet was re-suspended in 25 mL of isolation buffer, homogenized as previously described in isolation buffer and centrifuged at 700 g for 10 min. This second supernatant was mixed with the first and centrifuged at 7,000 g for 10 min to yield

a crude mitochondrial pellet. Half of each pellet was re-suspended in 10 mL of normal isolation buffer and half in isolation buffer containing 10 mM G6P at pH 6.3; both were then centrifuged at 7,000 g for 10 min. The resulting pellets were re-suspended in isolation buffer containing 25% (w/v) Percoll (pH 7.2 at 4 ˚C) and centrifuged at 17,000 g for 10 min. The mitochondrial pellets were re-suspended in isolation buffer and centrifuged again at 7,000 g for 10 min. The final purified mitochondrial pellets were re-suspended in a small volume of isolation buffer, and the protein concentration determined by the Biuret method using BSA as a standard. Mitochondria were kept on ice at a final concentration of 20–25 mg/mL and used within 4 hours. The protocol for mitochondrial preparation and treatment is summarized in S1 Fig.

For hearts used to assess the content of mitochondrial dynamic proteins (ie., Drp1, Mff and Dyn2) a modification of the polytron method [18] was used instead. Briefly, left ventricle was immersed in 6 mL ice-cold isolation buffer (300 mM sucrose, 3 mM EGTA, 10 mM Tris-HCl, pH 7.1) supplemented with 20 mM NEM, 1x complete protease inhibitors (Roche) and 1x phosphatase inhibitors. Tissue was rapidly chopped into fine pieces before homogenization using a Polytron tissue disruptor (Kinematica) at 10,000 rpm with 2 bursts of 5 seconds followed by 1 burst of 10 seconds. Tissue homogenate was diluted to 20 mL total volume with isolation buffer supplemented with 1x complete protease inhibitors and further homogenized by hand for 2 min using a glass Potter homogenizer and Teflon pestle. A small volume of homogenate was stored at -80 ˚C as whole homogenate. The homogenate was centrifuged at 7,500 g for 7 min. The pellet was rinsed twice with 5 mL isolation buffer, resuspended in 20 mL isolation buffer and further hand-homogenized for 2 min. The homogenate was then centrifuged at 600 g for 10 min and the supernatant was centrifuged at 7,000 g for 10 min to yield a crude mitochondrial pellet, which was resuspended in isolation buffer and stored at -80 ˚C. All fractions were assayed for protein concentration using a standard BCA assay protocol prior to use for Western blotting.

## Mitochondrial respiration

Oxygen consumption by isolated mitochondria was monitored polarographically with an Oxygraph-2k (Oroboros Instruments, Innsbruck, Austria) at 37 ˚C in a 2 mL chamber with standard respiration medium (KCl 125 mM, MOPS 20 mM, Tris 10 mM, EGTA 10 μM, $KH_2PO_4$ 2.5 mM, and $MgCl_2$ 2.5 mM, adjusted to pH 7.3 at room temperature with KOH). The respiratory substrates, 5 mM L-glutamate, 2 mM L-malate and 5 mM succinate, were added to the reaction chamber and allowed to equilibrate before addition of 0.25 mg/mL of mitochondria and closing of the chamber. After a stable state 2 respiration was achieved, ADP in excess (1.5 mM) was added, followed by 10 μM of cytochrome c to assess the permeability of the mitochondrial outer membrane to cytochrome c (indicated by the extent to which the rate of respiration was enhanced). Experiments were performed in duplicate using both Oxygraph-2k chambers. Respiration rates were calculated as the average value over a 45 s window in DatLab 5 (Oroboros Instruments) and are expressed in nmol $O_2$ / min /mg protein.

## Mitochondrial permeability transition

Opening of the mPTP in isolated mitochondrial fractions was assessed both in energized and de-energized conditions. In **energized conditions**, extramitochondrial [$Ca^{2+}$] was assayed by monitoring Fura-FF (Life Technologies, Paisley, UK) fluorescence in a multi-wavelength fluorimeter (Cairn Instruments, Kent, UK). Excitation filters, 340/20 nm and 380/20 nm, were contained in a spinning wheel rotating continuously at 32 Hz. Emission fluorescence was detected at 90˚ with a photomultiplier using a 520 nm bandpass filter. Mitochondria (0.25 mg/mL) were incubated at 30 ˚C within a stirred cuvette containing 2 mL assay buffer (KCl 125

mM, MOPS 20 mM, TRIS 10 mM, EGTA 20 μM, $KH_2PO_4$ 2 mM and Fura-FF 1 μM, pH 7.2 at 30 ˚C) supplemented with respiratory substrates (5 mM L-glutamate, 2 mM L-malate and 5 mM succinate). Additions of 20 μM free $[Ca^{2+}]$ were made every 2 min through an injection port (first pulse of 50 μM $[Ca^{2+}]_{total}$). Calcium-loading capacity was calculated as the sum of number of $Ca^{2+}$ pulses until mPTP opening, defined by an incomplete plateau before the next $Ca^{2+}$ addition.

Under **de-energized conditions**, mPTP was determined by following the decrease in light scattering at 520 nm ($A_{520}$) using a split-beam spectrophotometer with computerized data acquisition [27]. Mitochondria (0.25 mg/mL) were incubated at 25 ˚C within a stirred cuvette containing 2 mL assay buffer (KCl 125 mM, MOPS 20 mM, TRIS 10 mM, nitrilotriacetic acid (NTA) 2 mM, $KH_2PO_4$ 2.5 mM, rotenone 1 μM, antimycin A 1 μM and A23187 2 μM, pH 7.2 at 25 ˚C). A23187 was present to ensure complete equilibration of $Ca^{2+}$ across the mitochondrial inner membrane under de-energized conditions [28]. Addition of a single pulse of 83.5 μM $[Ca^{2+}]_{free}$ (1mM $[Ca^{2+}]_{total}$) was made through an injection port after an 80 s baseline. Variation in the initial $A_{520}$ (~0.5) was less than 10%. The initial rate of decrease in $A_{520}$ was calculated as the minimum of the first derivative of the $A_{520}$ time course.

In both cases, values of free $[Ca^{2+}]$ were calculated by a ligand-binding program (METLIG), as previously described [29]. Whenever indicated, 0.2 μM of cyclosporine A (CsA) was added in order to desensitize mPTP opening [30].

## Measurement of inner mitochondrial morphology

Extensive work from the laboratories of Sirak [31] and Klingenberg [32] demonstrated that ANT ligands modulate the light scattering (LS) of mitochondrial suspension as the ANT switches between its '*c*' and '*m*' conformations. Previous work from this laboratory has shown that these LS changes are independent of changes in mitochondrial matrix volume and may therefore provide an alternative means by which to monitor cristae morphology [33]. Here we use this approach in mitochondria isolated from the perfused heart either before ischemia of after 30 min ischemia with or without IPC.

**Light scattering measurements.** Light scattering (LS) was monitored at 520 nm ($A_{520}$) in a split-beam spectrophotometer with computerized data acquisition (1 data point per sec). Mitochondria (0.375 mg/mL) were suspended in assay buffer (KCl 125 mM, MOPS 20 mM, TRIS 10 mM, EGTA 0.5 mM, pH 7.2 at 25 ˚C) supplemented with respiratory substrates (5 mM L-glutamate, 2 mM L-malate and 5 mM succinate) and oligomycin (1 μM) at 25 ˚C. Equal volumes (2 mL) were used in sample and reference cuvettes and additions (463.5 μM $[Ca^{2+}]_{total}$ equivalent to 1 μM $[Ca^{2+}]_{free}$, 0.2 mM ADP and 5 μM CAT) were made to the sample cuvette as required through an injection port followed by stirring for 5 s. Variation in the initial $A_{520}$ (~0.5) was less than 10%. The magnitude of the changes in $A_{520}$ was calculated after signal stabilization and was expressed relative to the $A_{520}$ prior to effector addition (or baseline for the first addition). The rate of change in $A_{520}$ was calculated as the maximum (negative or positive) of the first derivative of the absorbance traces. Free $[Ca^{2+}]$ was calculated by a ligand-binding program (METLIG), as previously described [29].

**Transmission electron microscopy.** Assessment of mitochondrial ultrastructure was performed by TEM. Mitochondria were incubated as for LS measurements above, but protein concentration was scaled up to 0.75 mg/mL. When $A_{520}$ reached a constant signal, the mitochondrial suspension was fixed by addition of glutaraldehyde (3% (w/v) final) and incubated for 20 min at room temperature. Mitochondria were then sedimented by centrifugation at 3,000 g for 1 min and re-suspended in 0.1 M cacodylate for 10 min. The procedure was repeated twice before post-fixation with 1% (w/v) osmium tetraoxide in cacodylate buffer for

20 min at room temperature. Samples were washed 3 times with cacodylate buffer followed by standard embedding in Epon 812. Sections (60nm) were contrasted with Ur-acetate and Pb-citrate and analyzed on a FEI Technai12 TEM.

Mitochondria were classified and counted accordingly to their inner membrane morphology as illustrated in Fig 6: *Type I*, mitochondria present evenly spaced cristae with regular width; *Type II*, mitochondria present irregular cristae width but cristae are still thread-like, traversing from OMM to OMM. Spacing between cristae becomes irregular. *Type III*, mitochondria show a *Type II* configuration but at least two vesicular-like cristae are present. This reflects a combination between *Type II* and *Type IV* configurations. *Type IV*, mitochondria present rounded, nearly vesicular cristae without cristae traversing. Counting was performed in duplicate using non-overlapping micrographs and by two independent operators blind to the treatment whose scoring (agreeing within 10%) was averaged.

For morphometric analysis each mitochondrion was manually segmented and the respective mask used to measure its *area*, *perimeter*, *circularity* ($4\pi.[area/perimeter\verb|^|2]$) and *aspect ratio* (major axis/minor axis). For segmentation of all mitochondrial membranes a band pass filter was applied to the original EM picture in order to filter large structures down to 40 pixels and small structures up 10 pixels. Then, an Otsu threshold filtering was applied to the processed image and the resulting mask used to analyze particles with an average size bigger than 5000 pixels (S7 Fig). A summary of the data obtained is shown in the supplemental S4 Table. Image processing was performed in FIJI (ImageJ v1.51d).

## Western blot

Isolated mitochondrial fractions were diluted to 2.5 mg/mL in sample buffer (Tris 25 mM, glycerol 6% (w/v), SDS 5% (w/v), EDTA 1 mM and bromophenol blue 0.005% (w/v)) supplemented with 5% (w/v) β-mercaptoethanol and heated at 95 ˚C for 5 min. Proteins were separated in 5% (HK2 and HK1) or in 12% (ANT, Drp1, Mff, Dyn2) polyacrylamide casted gels by SDS-PAGE using 20 μg per lane. Gels were then subjected to Western blotting with the required primary antibody (HK2 (C64G5) rabbit monoclonal dilution 1:1,000, cat. #2867, Cell Signalling, Hertfordshire, UK; HK1 mouse monoclonal 5G9 dilution 1:1,000, cat. # WH0003098M1, Chemicon Millipore, Watford, UK; ANT—raised in-house against the C terminus of rat ANT [34], dilution 1:1,000; Drp1 mouse monoclonal dilution 1:1,000, cat. #611112, BD Biosciences, Wokingham; Dyn2 mouse monoclonal dilution 1:1,1000, cat. # SAB4200661, Sigma-Aldrich; Mff mouse monoclonal C11 dilution 1:2,000, cat. #sc-398731, Santa-Cruz Biotechnology, USA; GAPDH mouse monoclonal 6C5 dilution 1:20,000, cat. #ab8245, AbCam, Cambridge, UK; VDAC-1/2/3 rabbit polyclonal FL-283 dilution 1:1,1000, cat. #sc-9878, Santa-Cruz Biotechnology) and blots were developed using the required Ig HRP (horseradish peroxidase) secondary antibody, with ECL/ECL+ detection (Amersham Biosciences, Buckinghamshire, UK). Appropriate exposures of the film were used to ensure that band intensities were within the linear range. Quantification of blots was performed using an Alpha Inotech ChemiImager 4400 (Alpha Innotech Corp., San Leandro, CA, USA) to image the blot and analysis of band intensity with Quantity One (Bio-Rad Laboratories, Hercules, CA, USA). For densitometry, a rectangle with the maximum size similar to the band of greatest length present in the blot was considered as the region of interest and was used through the entire blot. Total band density, defined as the sum of all pixels intensity corrected for the local mean background was used. Each blot contained samples of (± low pH + G6P) pre-ischemic, end-ischemic and IPC end-ischemia mitochondria to allow direct comparisons between groups using the same film exposure. In order to normalize band intensities, parallel blots were performed on the same samples using antibodies against the ANT.

### Enzyme activities

Isolated mitochondrial fractions were disrupted using hypotonic shock induced by dilution to 2 mg/mL in phosphate buffer ($KH_2PO_4$ 33 mM pH 7.2 at room temperature) and incubation for 15 min on ice followed by one cycle of freeze-thaw.

Total **hexokinase activity** was evaluated in 1 mL of assay buffer (pH 7.4 at room temperature) containing Tris-HCl 100 mM, $NADP^+$ 0.4 mM, $MgCl_2$ 10 mM, ATP 5 mM, Triton X-100 0.3% (v/v), G6P dehydrogenase 0.5 U/mL and 60 μg of protein. The reaction was carried out at 37 ˚C in an Evolution 300 UV-Vis Spectrophotometer (Thermo Fisher Scientific, Waltham, MA, USA) and started after a 2 min baseline by addition of glucose (1 mM final). Hexokinase activity was calculated from the rate of NADPH production over the linear part of the assay (8 min) using an extinction coefficient of 6.22 $mM^{-1}.cm^{-1}$ [35]. Values were then normalized to citrate synthase activity and expressed as nmol NADPH/min/citrate synthase unit.

**Citrate synthase** activity was evaluated in 1 mL of assay buffer (pH 7.4 at room temperature) containing Tris-HCl 50 mM, Triton X-100 0.3% (v/v), and 5,5′-dithiobis-2-nitrobenzoic acid (DTNB) 150 μM, acetyl-coenzyme A 0.3 mM and 10 μg of protein. The reaction was carried at 37 ˚C and started after a 2 min baseline by addition of oxaloacetic acid (500 μM final). Citrate synthase activity was expressed in μmol /min/mg protein.

### Statistical analysis

All data were assessed for normality using the Kolmogorov-Smirnov test with Dallal-Wilkinson- Lillie for correction. Overall, analyzed data were normally distributed. Since group sample sizes are equal and the parametric statistical tests applied are robust for moderate deviations from homoscedasticity, parametric tests were always applied [36]. Statistical significance between groups was determined using matched-pairs (± G6P low pH) two-way ANOVA with interaction; *perfusion protocol* and *low pH + G6P* were the factors in use. Two types of comparisons were performed: 1) main *perfusion protocol* effect against *end-ischemia* group; 2) *low pH + G6P* effect individually analyzed for each perfusion group. In both cases, p values were adjusted for multiple comparisons through Holm-Šídák pos-hoc test and differences were considered significant at 5% level.

Frequency distributions of morphometric data from EM were expressed relatively to the total number of measured objects (mitochondria; S4 Table), with bin range and width defined automatically by the software (Graph Pad Prism). Afterwards, a Gaussian distribution was fitted to the data and independent fits between groups were compared to a global fit sharing the same *mean* by the extra sum-of-squares F test. The resulting p value was adjusted for multiple comparisons through Holm-Šídák pos-hoc test and differences were considered significant at 5% level.

Statistical analyses were performed using Graph Pad Prism version 6.01 (GraphPad Software, Inc., San Diego, CA, USA).

## Results

### Hemodynamic parameters during ischemia and IPC

Under the perfusion conditions used in the present studies, there were no differences in hemodynamic data between 'pre-ischemia' and 'end-ischemia' groups prior to entering ischemia (S1 Table). Hearts subjected to 30 min global ischemia showed an onset of ischemic contracture at 11.9 ± 0.7 min (S2 Table) and developed an infarct area upon 2 h reperfusion of 24.7 ± 2.0% (S3 Table). As expected, IPC treatment lowered aortic pressure and rate pressure product at the end of the stabilization period (S1 Table) and reduced the time to ischemic

contracture (7.8 ± 0.2 min) and infarct size (5.2 ± 0.7%) at reperfusion (S2 and S3 Tables, respectively).

### *In vitro* dissociation of mitochondrial-bound HK2

Previously we demonstrated that mitochondria-bound HK2 requires both an increase in G6P and a decrease in pH to be released from mitochondria *in situ* [18]. Thus, to promote HK2 dissociation *in vitro*, we modified the standard mitochondrial isolation protocol by including an extra washing step with either the standard isolation buffer (*normal*) or one supplemented with 10 mM G6P at pH 6.3 (*low pH + G6P*). It should be noted that the G6P and *low pH* are not present in the final wash and thus are not carried through into subsequent assays of mitochondrial function (see S1 Fig). We believe this protocol to be preferable to the use of synthetic peptides mimicking the HK2 binding peptide since these might be carried through into the assay and mimic the bound HK2 in its effects on the mPTP. Indeed, isolated cardiac mitochondria of mice overexpressing GFP fused to this peptide show successful dissociation of mt-HK2 but retention of the recombinant protein [37].

The western blot data of Fig 1a demonstrate that, as anticipated, our *low pH + G6P* protocol was effective in removing all bound HK2 from mitochondria isolated from control, ischemic and IPC ischemic hearts. These data also confirm our earlier observation that ischemia induces dissociation of mt-HK2 *in situ* which is prevented by IPC [18]. The fact that mt-HK2 from end-ischemic hearts can be further reduced *in vitro* indicates that the ischemic insult was not extreme, suggesting that further damage could have been attained to the heart. This agrees with our previous findings [13] where variations to the standard perfusion protocol led to different amounts of mt-HK2 being retained at end of ischemia, correlating with cardiac damage at reperfusion.

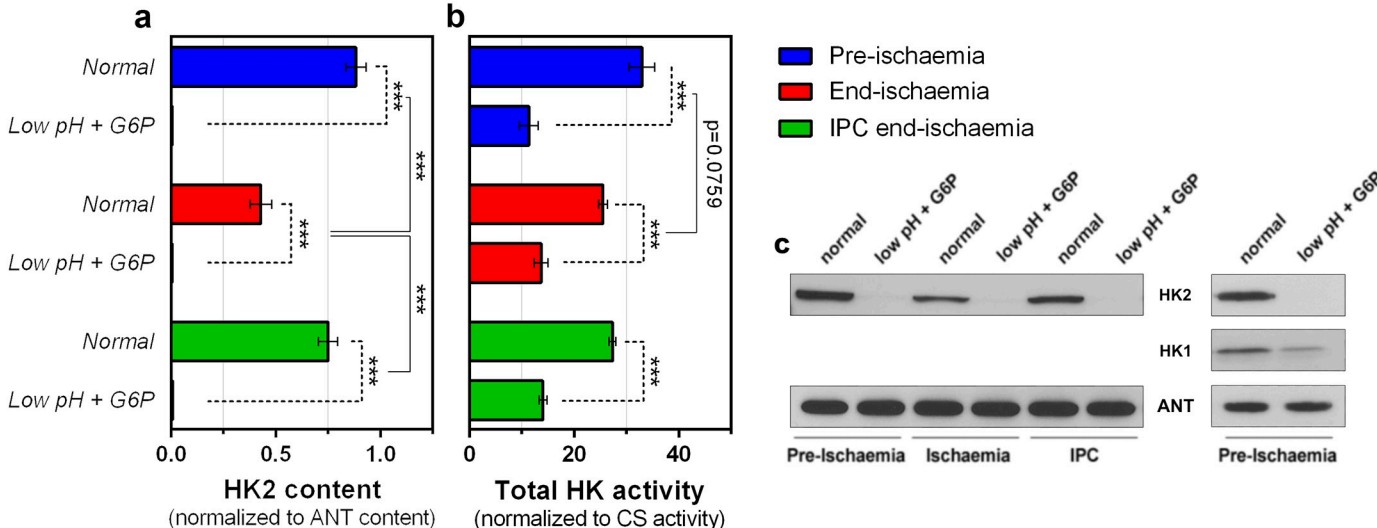

**Fig 1. Effects of ischemia and *low pH plus G6P* treatments on mitochondrial-bound HK2.** Incubation of mitochondria during isolation with factors know to induce HK2 dissociation [18] completely removes HK2 from mitochondria: a—HK2 protein content (representative blots are shown on panel c and uncropped membranes on S2b Fig), b—Total HK activity. Data are presented as mean ± SEM of 6–8 independent experiments on different mitochondrial preparations. Differences between groups were evaluated by a matched pair (± G6P low pH) two-way ANOVA with interaction followed by Holm-Šídák pos-hoc test to correct for multiple comparisons. $p_{interaction}^{HK2\ content} < 0.001$; $p_{interaction}^{HK\ activity} = 0.011$; ***, $p < 0.001$. Absolute values of citrate synthase (CS) activity (units per mg protein) were 2.30 ± 0.17, 2.59 ± 0.13, 2.69 ± 0.13 for pre-ischemia, end-ischemia and IPC end-ischemia, respectively. Corresponding values for low pH + G6P treated pre-ischemia, end-ischemia and IPC end-ischemia mitochondria were 2.32 ± 0.17, 2.38 ± 0.12 and 2.66 ± 0.18 respectively. c—representative blots for HK2 and HK1, as well as the ANT as loading control, of isolated mitochondrial fractions extracted from hearts perfused with different protocols (pre-ischemia, end-ischemia or IPC end-ischemia) ± *in vitro* HK2 dissociation protocol ('low pH plus G6P'); uncropped blots are shown on S2d Fig.

Further confirmation of the western blot data was provided by measurement of mitochondrial HK activity which was greatly reduced but not totally abolished by the protocol (Fig 1b). The remaining activity can be accounted for by the bound HK1 that was not dissociated by the treatment (Fig 1c), but which remains constant regardless of the perfusion protocol used as previously shown in *ex vivo* perfusion [13, 23]. It should be noted that low pH alone had no effect on total mt-HK activity (S2 Fig) or mt-HK2 content [18] in agreement with our previous data obtained in permeabilized cardiac fibers.

## *In vitro* dissociation of HK2 has no effect on mitochondrial respiration or outer membrane permeabilization

Next, we investigated the effects of mt-HK2 removal on mitochondrial respiration and OMMP as reflected by cytochrome c loss. Oxygen consumption was determined in the presence of both succinate that feeds electrons into Complex II and L-glutamate plus L-malate to generate NADH which feeds electrons into Complex I. These conditions most closely match the conditions *in vivo* [19]. Averaged traces (n = 8) of all analyzed groups are presented in Fig 2a and show that there is no significant difference between normal mitochondria and those depleted of HK2, regardless of the respiratory state (i.e. basal = state 2 and ADP-stimulated = state 3). Nevertheless, mitochondria from end-ischemic hearts exhibited an increase in state 2 respiration ($31.7 \pm 1.2$ *vs.* $25.6 \pm 1.6$ nmol $O_2$/min/mg of pre-ischemic, p = 0.004, n = 8) and depressed state 3 respiration (Fig 2b) as reported previously [19]. As expected, IPC treatment was able to prevent the latter, but state 2 rates remained slightly higher than pre-ischemic values ($34.65 \pm 1.4$ nmol $O_2$/min/mg, p = 0.056 *vs.* end-ischemia, n = 8). In order to quantify OMMP we measured the stimulation of state 3 respiration induced by addition of exogenous cytochrome c. Dissociation of HK2 alone had no effect on this parameter (Fig 2c), but as we have previously reported [18], end-ischemia mitochondria showed higher OMMP that was partially prevented by IPC treatment (Fig 2c). Overall our data suggests that HK2 dissociation alone is insufficient to account for the reported deleterious effects of ischemia on mitochondrial respiration and OMMP.

## *In vitro* HK2 dissociation does not sensitize mitochondria to mPTP opening

Another hallmark of ischemia-induced changes to mitochondria is an increased sensitivity to mPTP opening. We investigated the effects of ischemia, IPC and mt-HK2 dissociation on the sensitivity of mPTP opening to $[Ca^{2+}]$ under both de-energized and energized conditions. The rate of $A_{520}$ decrease induced by addition of $Ca^{2+}$ was used to determine the extent of mPTP opening under de-energized conditions. Average traces (n = 5–6) in the absence and presence of CsA are shown in Fig 3a and S3 Fig respectively, while mean rates of swelling ($\pm$ SEM) are reported in Fig 3b. Addition of a single bolus of $Ca^{2+}$ to end-ischemia mitochondria caused a fast and large decrease in $A_{520}$ that was not observed in pre-ischemia mitochondria, confirming previous data [38] that ischemia enhances the sensitivity of the mPTP to $[Ca^{2+}]$ and that IPC treatment was able to protect against this sensitization. As expected, incubation with CsA inhibited the mPTP regardless of the perfusion protocol (Fig 3b), but differences between groups were still observed. However, when HK2-depleted mitochondria were analyzed, no differences were observed relative to their matched controls, with one exception. The rate of $A_{520}$ change was slightly lower in HK2-depleted end-ischemia mitochondria compared to normal end-ischemia mitochondria, although the response to CsA was similar in both groups.

The sensitivity of mPTP opening to $[Ca^{2+}]$ was also evaluated under energized conditions using the calcium retention capacity assay. Average traces (n = 4–6) in the absence and

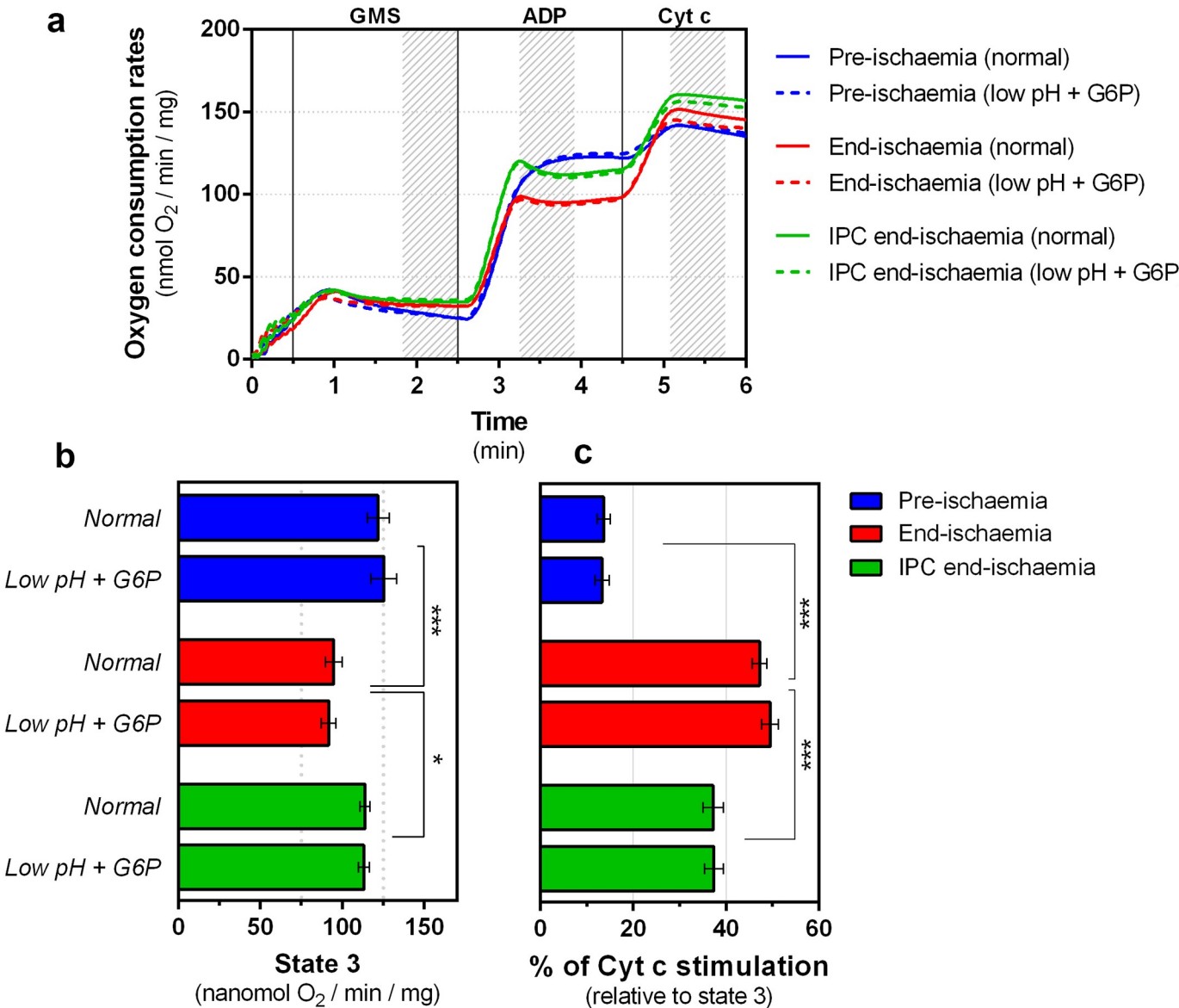

**Fig 2. Effects of ischemia and *low pH plus G6P* treatments on mitochondrial respiration.** Traces (a) are mean data of 8 independent experiments which are analysed further in panels b-c where data are presented as means ± SEM. The shadow area on the top graph (a) represents the range used to calculate respiration rates. (b) State 3 represents respiration in the presence of ADP. (c) Indirect assessment of OMMP by cytochrome c stimulation of mitochondrial respiration. Differences between groups were evaluated by a matched pair (± G6P low pH) two-way ANOVA with interaction followed by Holm-Šídák pos-hoc test to correct for multiple comparisons. *, p < 0.05; ***, p < 0.001.

presence of CsA are shown in Fig 4a and S4 Fig, respectively, while mean data (± SEM) for the calcium loading capacity are reported in Fig 4b. As expected, end-ischemia mitochondria accumulated less calcium compared to pre-ischemic mitochondria suggesting a higher sensitivity to pore opening, and IPC treatment attenuated this effect. CsA also attenuated mPTP opening but differences between groups were still present as was also observed under de-energized conditions. However, just as observed under de-energized conditions, the sensitivity of HK2-depleted mitochondria to mPTP opening was similar to control-matched mitochondria, regardless of the perfusion protocol or the presence of CsA. Taken together, these data imply

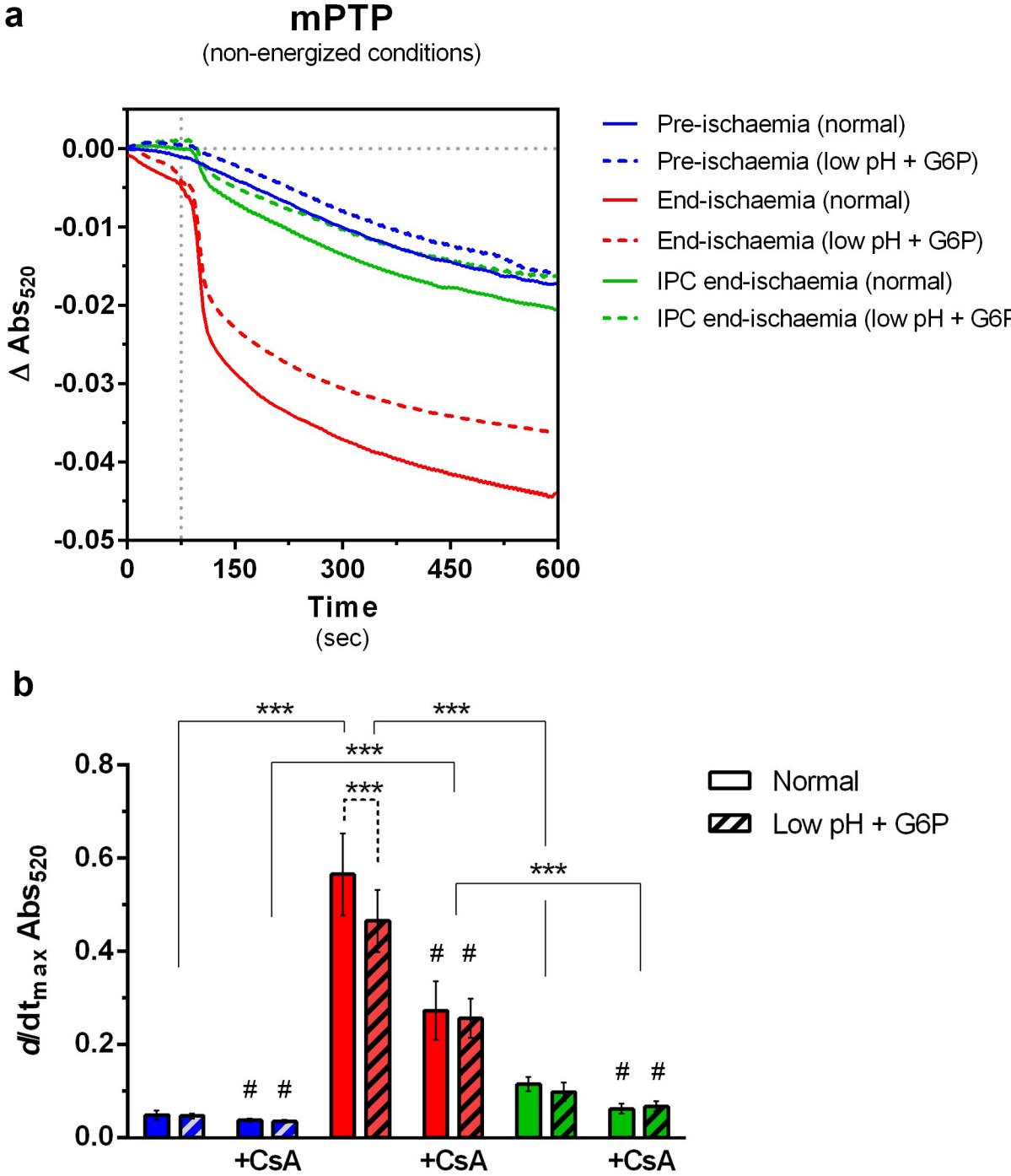

**Fig 3. Effects of ischemia and *low pH plus G6P* treatments on mPTP opening under de-energized conditions.** (a) Mitochondrial swelling associated with mPTP opening was triggered by addition of 83.5 μM free $[Ca^{2+}]$ and monitored by measurement of $A_{520}$. Traces (a) are mean data of 5–6 independent experiments which are analysed further in panels b-c where data are presented as means ± SEM. (b) The initial swelling rate ($d/dt_{max} A_{520}$) was calculated as the minimum of the first derivative of the absorbance traces and is shown as an absolute value. Cyclosporin A (CsA—0.2 μM) was used to inhibit mPTP opening. Individual and averaged traces are presented in S3 Fig. Differences between groups were evaluated by a matched pair (± G6P low pH) two-way ANOVA with interaction followed by Holm-Šídák pos-hoc test to correct for multiple comparisons. $P_{interaction}^{without\ CsA} = 0.002$. ***, $p < 0.001$. , $p < 0.05$ vs matched group without CsA.

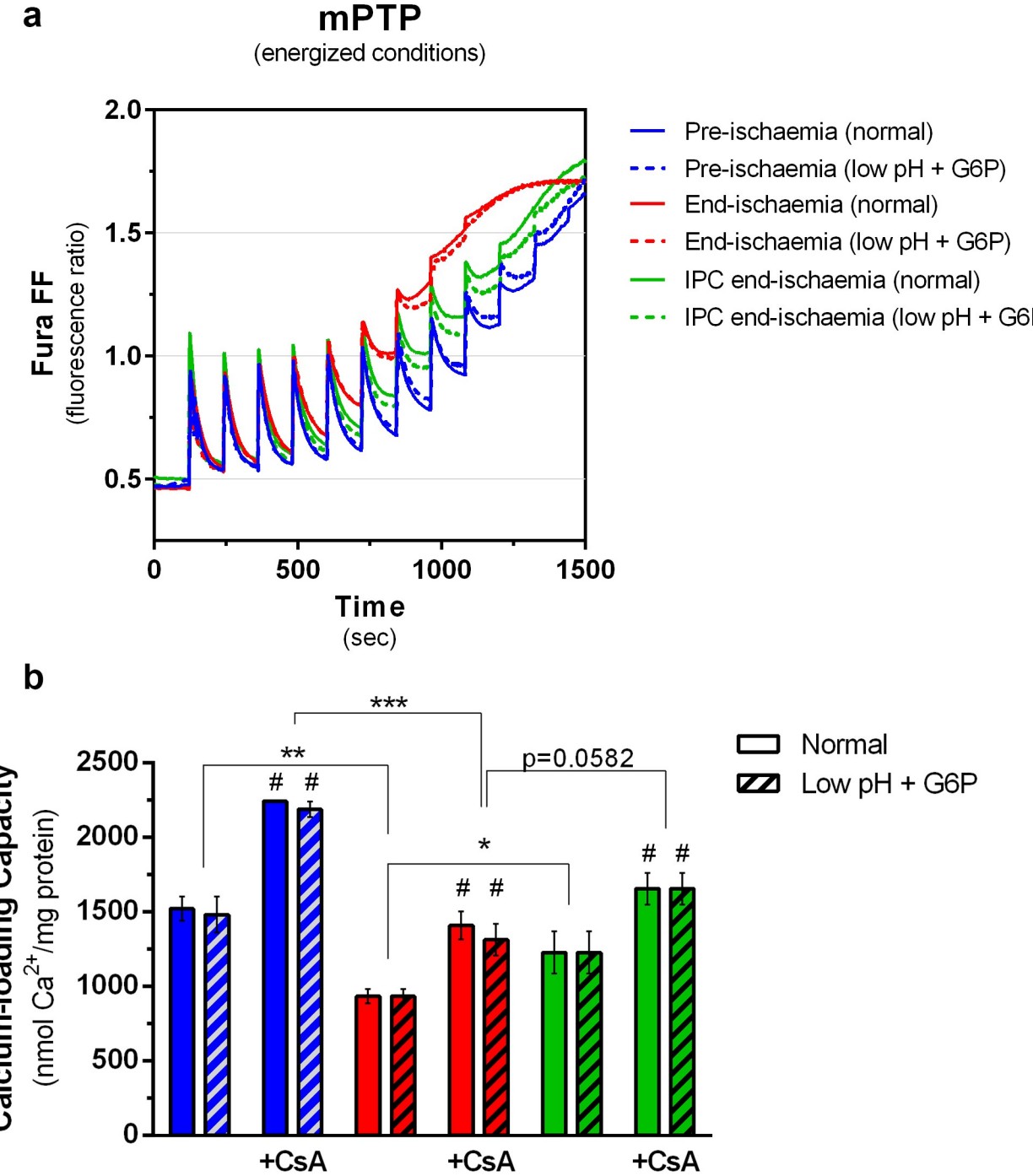

**Fig 4. Effects of ischemia and *low pH plus G6P* treatments on the sensitivity to mPTP opening under energized conditions.** (a) mPTP opening was evaluated by the calcium retention capacity assay with addition of 20 μM free $[Ca^{2+}]$ every 2 min. (b) Calcium-loading capacity was calculated as the sum of the number of $Ca^{2+}$ pulses until mPTP opening, defined by an incomplete plateau before the next $Ca^{2+}$ addition. Traces (a) are mean data of 4–6 independent experiments which are analysed further in panels b-c where data are presented as means ± SEM. CsA (0.2 μM) was used to inhibit mPTP opening and, individual and averaged traces can be found in S4 Fig. Differences between groups were evaluated by a matched pair (± G6P low pH) two-way ANOVA with interaction followed by Holm-Šídák pos-hoc test to correct for multiple comparisons. *, $p < 0.05$; **, $p < 0.01$; ***, $p < 0.001$. #, $p < 0.05$ vs matched group without CsA.

that the increased sensitivity to mPTP opening observed after ischemia, like the changes in outer membrane permeability and respiration, must occur downstream of HK2 dissociation from mitochondria. We postulated that this might involve proteins such as Drp1 that are involved in mitochondrial fusion / fission and which we have previously proposed may provide a link between HK2 binding and modulation of mPTP activity [14].

### Ischemia promotes Drp1 translocation to mitochondria *in vivo* but this is not prevented by IPC

Drp1 translocation from cytosol to mitochondria is known to occur upon reperfusion [39] but less is known about its localization during ischemia. In the light of the reported effects of Drp1 on the mPTP [40, 41] and the cardioprotection afforded by its inhibitor mdivi-1 [42], we investigated Drp1 association with mitochondria at the end of ischemia. In Fig 5 we demonstrate that the levels of the mitochondrial fission receptor (Mff), which is responsible for Drp1 recruitment, were significantly lower in end-ischemia mitochondria compared to the pre-ischemic control (Fig 5, left). However, the amount of Drp1 present in end-ischemia

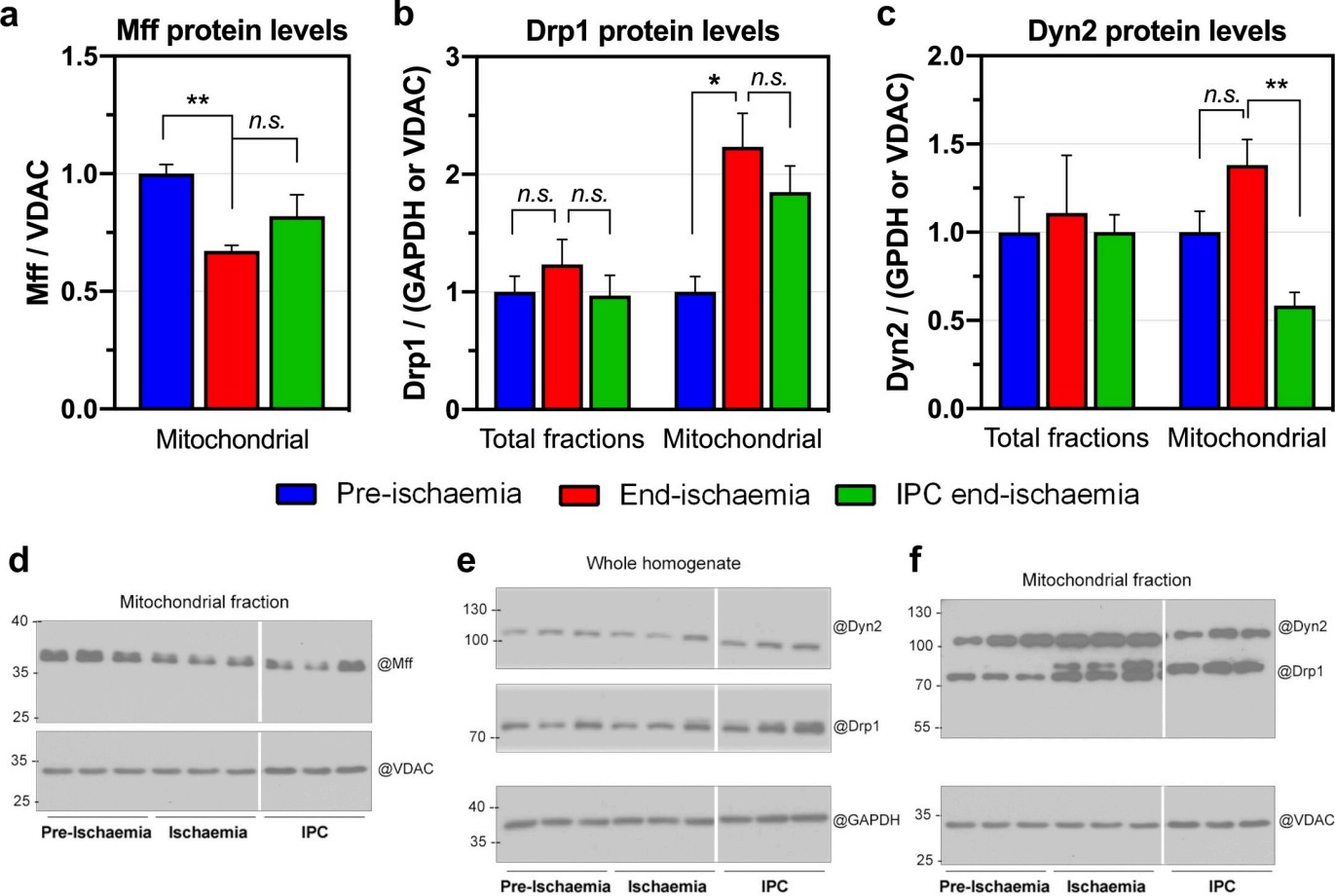

**Fig 5. Effects of ischemia on the levels of mitochondrial fission proteins.** The indicated proteins were collected either from total heart extracts or isolated mitochondrial fractions and resolved by SDS-PAGE followed by immunoblotting (for uncropped blots see S5 Fig). Ischemic damage was assessed indirectly by evaluating time to ischemic contracture [13]: 13.0 ± 0.8 min *vs*. 7.7 ± 1.2 min, p = 0.004, end-ischemia *vs*. IPC, respectively. Data in graphs are shown as mean ± SEM of 6 independent experiments. Differences between groups were run separately for 'total' and 'mitochondrial' datasets and were evaluated by one-way ANOVA followed by Dunnett's post-hoc test to correct for multiple comparisons. *, p < 0.05; **, p < 0.01.

mitochondria was higher than in their control counterparts (Fig 5, middle), suggesting that Drp1 may be recruited to sites other than Mff. IPC prevented neither the increase in mitochondrial Drp1 nor the decrease in Mff associated with mitochondria at the end of ischemia. These data imply that cardioprotection by IPC is independent of changes in Drp1 recruitment but this may not rule out an effect of IPC on the mitochondrial fission machinery since it has been shown recently that dynamin-2 (Dyn2) acts downstream of Drp1 recruitment in order to complete fission of mitochondrial membranes [43]. Thus, we measured the amount of Dyn2 levels present in the mitochondria. In contrast to Drp1, the amount of mitochondria-associated Dyn2 at the end of ischemia was only slightly higher than pre-ischemic mitochondria (Fig 5, right; p = 0.060), but IPC significantly decreased mitochondria-associated Dyn2 (Fig 5, right). This reduction reflects an effect of IPC on the recruitment Dyn2 to mitochondria since cellular Dyn2 levels remained unchanged (Fig 5, right, 'total').

Overall, our data show that although Drp1 recruitment alone cannot explain the increased sensitivity to mPTP observed after ischemia, it remains possible that it is the combination of Drp1 and Dyn2 associated with mitochondria that can regulate the sensitivity of the mPTP through changes in mitochondrial morphology [44]. In light of these observations, we investigated the effects of ischemia on IMM morphology.

## Effects of ischemia on inner mitochondrial morphology

**Validating the use of light scattering to detect changes in mitochondrial morphology.** Previous work in this laboratory [27, 33] has shown that small changes in light-scattering (LS) of isolated heart mitochondria can be induced by ligands of the adenine nucleotide translocase (ANT) or sub-micromolar [$Ca^{2+}$]. These LS changes are much smaller than those accompanying mPTP opening and occur in the absence of significant alterations in total organelle volume. Although it remains unclear how the conformation of the ANT leads to changes in LS, it is thought that it reflects changes in the inner mitochondrial morphology. The data of Fig 6 confirm this by using transmission electron microscopy (TEM) to evaluate mitochondrial ultrastructure in parallel with LS measurements with the intention of using LS as a quantitative measure of any small changes in the inner membrane morphology that may accompany ischemia and HK2 loss. In these experiments we used $Ca^{2+}$ and two ligands of the ANT, ADP and carboxyatractyloside (CAT), to modulate LS. ADP was chosen instead of bongkrekic acid because the latter binds irreversibly to the ANT whereas the effects of ADP can be reversed by CAT. Importantly, the effect of ADP is independent of ATP synthase activity since the buffer is lacking phosphate (Pi) and is supplemented with oligomycin (a specific ATP synthase inhibitor). We also confirmed that the ANT content of mitochondria was unaffected by ischemia and IPC (S6 Fig) and thus could not account for any changes in the effects of ANT ligands seen between mitochondria following these treatments.

Fig 6a shows a typical trace of such a LS experiment for pre-ischemia mitochondria. As expected, addition of a small amount of $Ca^{2+}$ (1 μM free [$Ca^{2+}$]), which is not sufficient to induce mPTP opening, caused a decrease in LS that could be reversed by addition of 0.2 mM ADP to induce the 'm' conformation of the ANT. The effect of ADP was itself rapidly reversed by addition of 5 μM CAT (Fig 6a). In Fig 6b we use TEM to confirm that the $Ca^{2+}$-induced effects in LS are associated with changes in IMM morphology rather than total volume changes (Fig 6c). This conclusion assumes that mitochondria maintain a spherical shape after isolation, as reported by others [45]. We observed that IMM morphology could be broadly classified into four distinct categories, illustrated in Fig 6b. $Ca^{2+}$ addition induced a clear transition from a *classical* conformation (Type I) towards rounded, nearly vesicular cristae without cristae traversing, ie. Type III-IV ($p_{Type\ I}$ < 0.001 and $p_{Type\ III}$ < 0.001, vs. baseline; Fig 6B). In

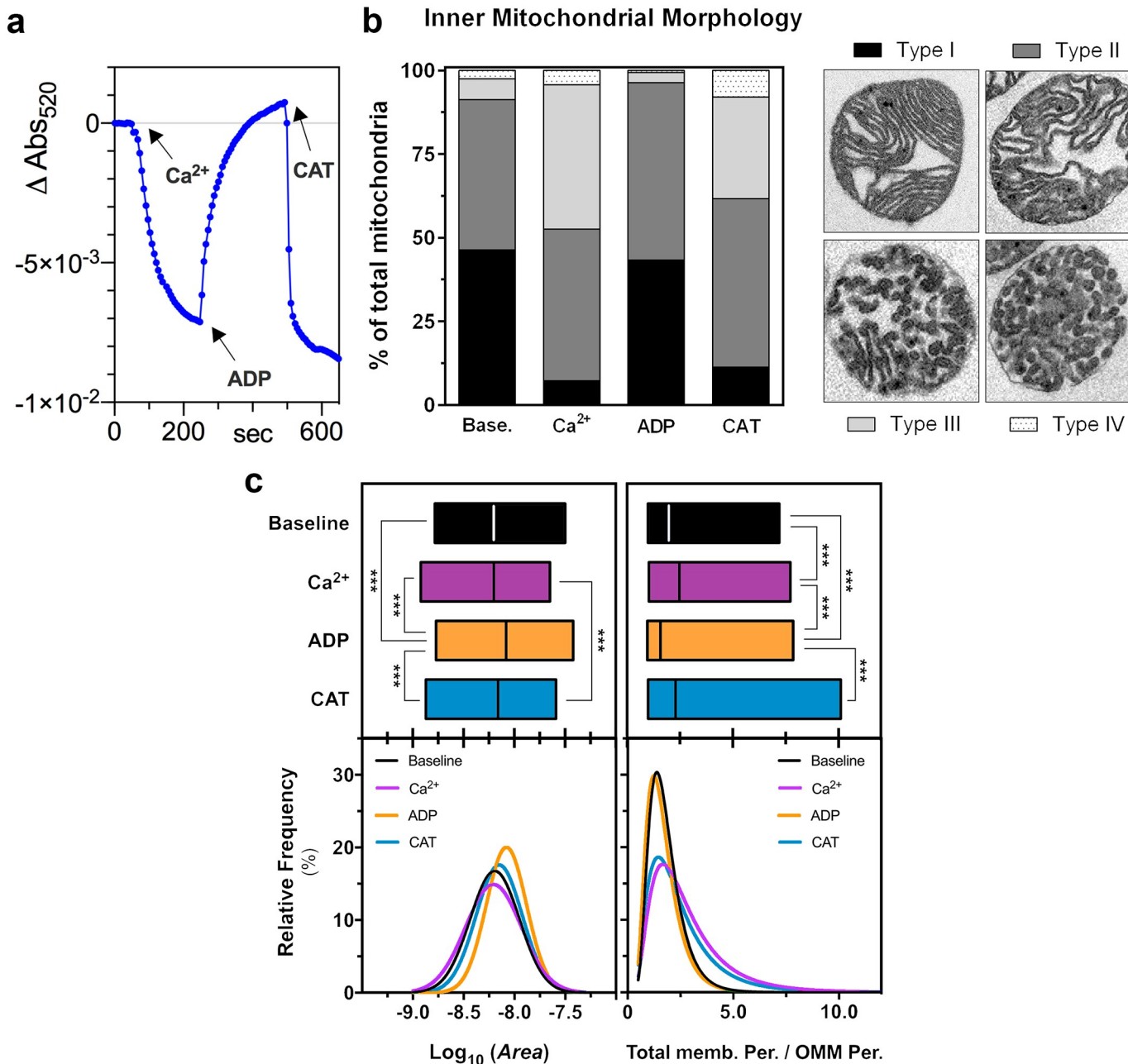

**Fig 6. Effects of ANT ligands on the inner mitochondrial morphology in pre-ischemic mitochondria.** a—monitoring of small fluctuations in light scattering in the presence of ANT ligands in pre-ischemia control mitochondria. b—A sub-set of samples of the light scattering experiments shown in (a) were chemically fixed and analyzed by TEM. Mitochondria were classified and counted accordingly to their inner membrane morphology by two independent operators blind to the treatment. *Type I*, mitochondria present evenly spaced cristae with regular width, ie. classical conformation; *Type II*, mitochondria present irregular cristae width but cristae are still thread-like, traversing from OMM to OMM. Spacing between cristae becomes irregular. *Type III*, mitochondria show a *Type II* configuration but at least two vesicular-like cristae are present. It reflects a combination between *Type II* and *Type IV* configurations. *Type IV*, mitochondria present rounded, nearly vesicular cristae without cristae traversing. Data in bars are presented as means of 2 independent experiments, each scored by two independent observers. Total number of mitochondria employed (193–347) and other morphometric parameters can be found on S4 Table. c—frequency distributions of morphometric data from TEM cross sections expressed relatively to the total number of analyzed mitochondria. Details about 'OMM perimeter' and 'Total membrane perimeter' can be found in Material and Methods and in the Supplemental Material. A Gaussian distribution was fitted to the data and independent fits between groups were compared to a global fit sharing the same *mean* by the extra sum-of-squares F test and p values adjusted for multiple comparisons through Holm-Šídák pos-hoc test. ***, $p < 0.001$. Abbreviations: Base.–baseline, CAT—after carboxyatractyloside treatment, end-isch.—end-ischemia, Per.—perimeter.

addition to investigate IMM structures further, morphometric analysis of EM cross sections of the mitochondria was performed by determining the distribution of total membrane perimeter (including IMM, calculated as described in Material and Methods and S7 Fig) normalized to the OMM perimeter (Fig 6c, $Ca^{2+}$ vs. baseline). These data show a shift towards higher values, in agreement with an IMM rearrangement towards vesicular cristae.

When ADP was added after $Ca^{2+}$ to reverse the decrease in LS, TEM micrographs confirmed that this was accompanied by restoration of evenly spaced cristae with regular width, i.e. Type I (Fig 6b; $p_{Type\ I}$ and $p_{Type\ III}$ > 0.05, vs. baseline). Interestingly, morphometric analysis showed that mitochondria presented a larger area in the presence of ADP (Fig 6c, left), but assuming a spherical shape upon isolation, one should expect a decrease in LS as a consequence of higher cross-section area. This is the opposite of what is observed (Fig 6a) and suggests that morphological changes have a stronger effect on LS compared to total volume changes, under these experimental conditions. Therefore, these observations indirectly confirm the robustness of our optical approach to monitor changes in inner membrane morphology. The full recruitment of ANT to its 'c' conformation was achieved by addition of CAT. This caused a rapid drop in LS (Fig 6a) which was associated with a change in the inner membrane morphology towards Type III-IV (Fig 6b; $p_{Type\ I}$ < 0.001 and $p_{Type\ III}$ = 0.013, vs. baseline). As might be expected, the total area and total membrane perimeter distributions more closely resemble those in the presence of $Ca^{2+}$, in agreement with the IMM classification. Taken together these data confirm that under our experimental conditions, alterations in the IMM morphology induced by ligand-induced changes in ANT conformation are detected as small changes in LS as concluded previously [27, 33].

**Effect of ischemia and HK2 dissociation on mitochondrial ultrastructure.** In Fig 7 we present TEM data on the effects of ischemia on IMM morphology. To our surprise, end-ischemia mitochondria were observationally no different from pre-ischemic mitochondria and were mainly classified as Type I mitochondria (*classical*; Fig 7a). However, morphometric analysis did reveal that end-ischemia mitochondria were slightly, but significantly larger than pre-ischemic mitochondria (Fig 7b), consistent with a greater matrix volume as we have reported previously [46].

To explore any subtle differences in mitochondrial morphology induced by ischemia and IPC that might reflect changes in inner and outer membrane interactions or cristae structure that are not readily visible by TEM, we investigated the response of IMM to $[Ca^{2+}]$ and ANT ligands using the LS technique as described in Fig 6. This technique is also more readily quantifiable and much less labor intensive than using TEM (Fig 8a). The amplitude and rate change of LS induced by $Ca^{2+}$ was considerably larger and faster in mitochondria from pre-ischemic hearts than those from end-ischemia hearts (Fig 8b and 8e). However, IPC treatment failed to re-establish the $Ca^{2+}$-induced drop in LS to control values (Fig 8b). When ADP was added after $Ca^{2+}$, the amplitude and rate change in LS were consistently higher in mitochondria from pre-ischemic hearts than those from end-ischemia hearts (Fig 8c and 8f). Interestingly, although the LS plateau achieved after $Ca^{2+}$ addition by end-ischemia and IPC end-ischemia was similar (Fig 8a), IPC promoted a larger and faster ADP-induced change in LS compared to end-ischemia group (Fig 8c and 8f). Similarly, the amplitude and rate of CAT-induced change in LS were consistently lower in end-ischemia mitochondria compared to pre-ischemia mitochondria (Fig 8d and 8g).

Next, we investigated the LS responses of mitochondria depleted of HK2 by prior treatment with G-6-P at pH 6.3. The overall behavior of the HK-2 depleted mitochondria was similar to their control counterparts with the following exceptions (Fig 8a, dashed lines). First, the amplitude of the $Ca^{2+}$-induced decrease in LS of HK2-depleted pre-ischemia mitochondria was lower than for un-depleted controls, but still higher than end-ischemia mitochondria (Fig 8b).

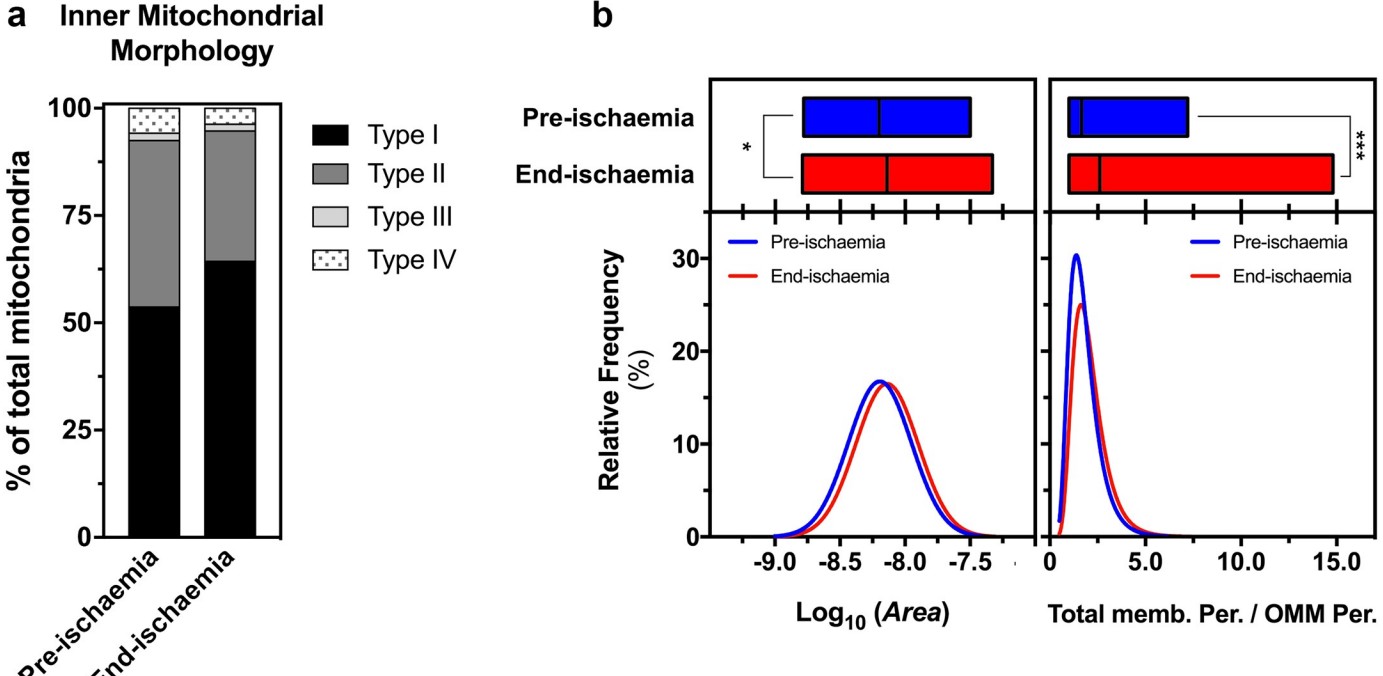

**Fig 7. Inner mitochondrial classification of pre-ischemia *vs.* end-ischemia mitochondria (a) and the corresponding morphometric analysis of EM cross sections (b).** Data were processed as described in Fig 6 and are shown as mean ± SEM of 2 independent experiments. Total number of mitochondria employed (193 for pre-ischemia and 412 for end-ischemia). *, p < 0.05; ***, p < 0.001.

Second, the rate of ADP-induced LS change of HK2-depleted IPC end-ischemia mitochondria was slower than for the same mitochondria without HK2-depletion, but still higher than end-ischemia mitochondria (Fig 8f).

Overall, our data reveal that ischemia, which decreases mt-HK2 binding, decreased the LS response to ANT ligands while IPC restored these effects. This implies that in parallel with its depletion of mt-HK2, ischemia reduces the coupling between ANT conformation and IMM, perhaps reflecting changes in inner and outer membrane interactions or cristae structure that are not readily visible by TEM, and that this effect is prevented by IPC. However, just as with the other parameters studied, modulating mt-HK2 alone had only a minor effect on this coupling, suggesting that events other than mt-HK2 dissociation occur *in situ* to prime mitochondria for mPTP opening on reperfusion.

## Discussion

The role of mt-HK2 in cardioprotection has been extensively studied in this [13, 18, 19] and other [20–23] laboratories. Pre-ischemic interventions, such as IPC, which prevent mt-HK2 dissociation afford substantial cardioprotection [13]. Indeed, we found a strong inverse correlation between the amount of mt-HK2 remaining at end of ischemia and the infarct size upon subsequent reperfusion. Although these studies have identified ischemia-induced events upstream and downstream of HK2 displacement, it has not been established whether mt-HK2 dissociation is the primary signal mediating the ischemia-induced changes to mitochondria that enhance mPTP opening on reperfusion. In the present investigation we have applied a battery of biochemical techniques and ultrastructure imaging tools to assess whether mt-HK2

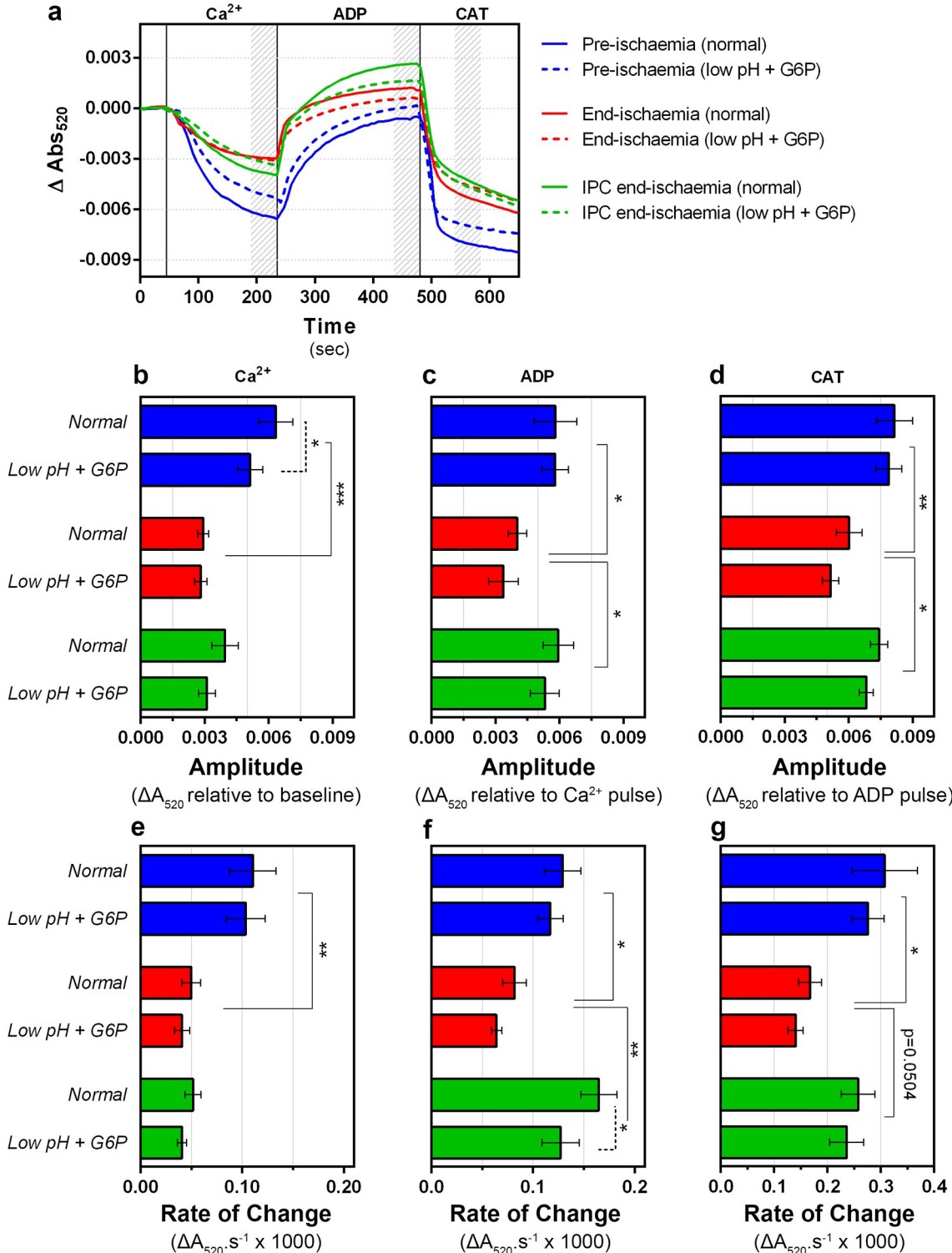

**Fig 8. Effects of ischemia and *low pH plus G6P* treatments on the inner mitochondrial morphology.** (a) Inner mitochondrial morphology was assessed indirectly by following small fluctuations in light scattering in the presence of ANT ligands. Traces (a) are mean data of 8 independent experiments which are analysed further in panels b-g where data are presented as means ± SEM. (b-d) The shadow area on the top graph represents the range used to calculate the amplitude of changes in $A_{520}$. Amplitude values were calculated as the absolute difference between the final averaged value after the previous addition (or baseline if the first) and the last averaged value of the current addition. (e-g) Rates of change in $A_{520}$ ($d/dt\,A_{520}$) were calculated as the maximum (negative or positive)

of the first derivative of the absorbance traces. Differences between groups were evaluated by a matched pair (± G6P low pH) two-way ANOVA with interaction followed by Holm-Šídák pos-hoc test to correct for multiple comparisons. *, p < 0.05; **, p < 0.01; ***, p < 0.001.

dissociation alone is sufficient to reproduce some or all of the changes in mitochondrial function observed following ischemia. Our rationale is that, if prevention of HK2 dissociation from mitochondria is directly responsible for the attenuation of changes to mitochondria function at the end of ischemia, then its removal from IPC end-ischemic mitochondria should change these parameters back to those of non-preconditioned end-ischemic mitochondria. As in our previous work with permeabilized fibres [18], we used the combination of high [G6P] with low pH (6.3) to dissociate mt-HK2 *in vitro* during isolation of mitochondria which were then returned to normal isolation buffer to ensure that all assays were performed under identical conditions. This approach, which mimics ischemic conditions and is specific for HK isoform 2 (Fig 1), has advantages over peptide- or drug-based approaches, such as TAT-HK, AnP-HK or clotrimazole, that can exert non-specific effects [17, 18, 47, 48]. For example, in the case of synthetic peptides mimicking the HK2 binding peptide these might be carried through into the assay and mimic the bound HK2 in its effects on the mPTP. Indeed, isolated cardiac mitochondria of mice overexpressing GFP fused to this peptide show successful dissociation of mt-HK2 but retention of the recombinant protein [37].

It is important to note that our experimental design using isolated mitochondria allows us to make important observations which would extremely difficult, if not impossible, to perform using in intact cells / in vivo. First, our protocol dissociates HK2 from the mitochondria once isolated thus ensuring the absence of cytoplasmic factors that might exert effects on mitochondrial function or morphology that are secondary to mt-HK2 release. Second, we isolate mitochondria either from baseline or after a damaging insult ± protective regimen, allowing us to detect intramitochondrial differences that could account for increased sensitivity to pore opening upon mt-HK2 release since changes would be maintained in our experimental setup. In this context, we have previously shown [7, 19] that mitochondria isolated at end-ischaemia ± IPC show similar $[Ca^{2+}]_m$ accumulation while ischemic mitochondria show increased ROS production, which are well established mPTP triggers. The fact that complete removal of mt-HK2 does not change the sensitivity of the mPTP either in the baseline or the other tested groups, which have mitochondria with modified behaviour, suggests that mt-HK2 was not acting as "the pin of a grenade".

## Loss of mt-HK2 binding is not the primary cause of OMM permeabilization, cytochrome c loss and mPTP sensitization during ischemia

If it were the dissociation of mt-HK2 from mitochondria during ischemia that were directly responsible for cristae widening, OMMP, cytochrome c loss and mPTP sensitization during ischemia, it would be anticipated that removal of mt-HK2 *in vitro* with high [G6P] and low pH would mimic the effect of ischemia. However, in Fig 2 we demonstrate that mt-HK2 dissociation on its own is not sufficient to induce OMMP and cytochrome *c* release. This implies that the entities responsible for OMMP are not pre-inserted in the mitochondria during ischemia such that they could initiate their action upon HK2 release. Indeed the only member of the Bcl2 family of proteins regulating OMMP that is changed in mitochondria following ischemia is Bcl-XL whose expression decreases [19], but IPC did not prevent this loss despite preventing

cytochrome c release [18]. Thus, we propose that the ability of IPC to maintain mitochondrial morphology and OMM integrity during ischemia, and so facilitate cytochrome c retention, is not a direct effect of preventing mt-HK2 dissociation.

Similarly, *in vitro* dissociation of mt-HK2 from mitochondria did not sensitize them to mPTP opening (Fig 5) which is in stark contrast to the sensitization to mPTP opening seen in mitochondria from ischemic hearts, where mtHK2 dissociation also occurs. The fact that removal of mt-HK2 from IPC mitochondria does not increase mPTP activity to the level observed in end-ischemia mitochondria provides evidence that the IPC-mediated inhibition of mPTP opening is not caused directly by the bound HK2. This was somewhat unexpected in the light of published data on the role of HK in apoptosis [16, 49] and mPTP modulation [17] and implies that there is another regulatory factor involved in the sensitization of the mPTP by ischemia that is attenuated by IPC. This is unlikely to involve either calcium or ROS since we have shown that IPC modulates neither of these parameters in end-ischemic mitochondria [7] which led us to focus our attention on Drp1, whose association with I/R injury has been studied previously [39].

## Drp1 recruitment during ischemia alone cannot explain mitochondrial sensitization to I/R injury

It has been reported that an increase in the intermembrane space occurs during mitochondrial remodeling in apoptosis and that this is dependent on Dynamin-related protein 1 (Drp1) [50]. Drp1 is a cytosolic protein that translocates to mitochondria to initiate fission [43]. Furthermore, a similar redistribution of Drp1 to mitochondria has previously been reported to occur upon reperfusion after ischemia [39]. These data are consistent with mitochondrial remodeling playing an important role in determining the outcome of I/R damage. An attractive hypothesis that we have discussed in more detail previously [14], is that mt-HK2 binding might prevent the association of cytosolic factors, such as Drp1, with mitochondria during ischemia and that it is these proteins that are the direct cause of the ischemic changes to mitochondria that activate the mPTP upon reperfusion. This would be consistent with the ability of the pharmacological inhibitors of Drp1, mdivi-1 (GTPase activity) and the peptide inhibitor P110 (Drp1-Fis1 interaction) to reduce infarct size in various models of cardiac I/R [42, 51], even though side-effects [52] and target requirements [53] are disputed. However, although we confirmed that ischemia did increase the recruitment of Drp1 to mitochondria, IPC failed to prevent this despite affording substantial cardioprotection (Fig 5). Thus, Drp1 binding to mitochondria alone cannot be responsible for ischemia-induced changes to mitochondria that prime them for mPTP opening on reperfusion. However, it should be noted that Drp1 activity and binding to mitochondria can be modulated by phosphorylation at various sites, as well as by other post-translational modifications such as sumoylation and nitrosylation, although there is considerable controversy over the effects, role and relevance of these modifications [40, 54, 55]. Furthermore, only modifications that inhibit the fission activity of Drp1 without causing its dissociation from mitochondria could account for our results and there is no clear evidence for such a mechanism. In addition, previous data from this laboratory have provided no evidence for a mitochondria-associated phosphorylated protein of the right size for Drp1 [11]. Another factor to consider is that Drp1 translocation is just one step in a cascade of events that occur in order to constrict and allow polymeric ring-like structures to severe mitochondria [43]. The process starts with ER-mediated constriction of mitochondria at pre-Drp1 sites in an Mff- and MiD49/51-independent manner, followed by Drp1 recruitment by its adaptor proteins and finishes with Dyn2 recruitment and assembly to complete fission [43]. Interestingly, ablation of Mfn2, a mitochondrial adaptor protein necessary for ER-

mitochondria tethering, increases infarct size [56]. Indeed, modulation of any of these check-points [51, 57, 58], or just inhibition of the last step, such as inhibition of Dyn2 by dynasore [59], is sufficient to attenuate I/R injury. Our data from IPC treatment suggests that mt-HK2 cardioprotective mechanism could be due, in part, to prevention of mitochondrial recruitment of Dyn2 rather than Drp1. However, the relationship between HK2 binding to mitochondria and changes in Drp1/Dyn2 recruitment remains unclear. It is unlikely that mt-HK2 binding to the OMM masks sites that bind Drp1 since, if this were the case IPC, which reduces mt-HK2 loss, should prevent the recruitment of Drp1 during ischemia which it did not. Nevertheless, our data would be consistent with mt-HK2 attenuating the binding of Dyn2 that follows recruitment of Drp1. We cannot rule out the possibility that the changes in mt-HK2 are sec-ondary to changes in the recruitment of both Drp1 and Dyn2 and that mt-HK2 plays no role in modulating mitochondrial morphology and mPTP sensitivity during ischemia. However, our IMM morphometric analysis, discussed further below, suggests that contact sites between the inner and outer membranes, which are disrupted in ischemia, may be involved in modulat-ing mPTP sensitivity and hence in cardioprotection. Thus, dissociation of mt-HK2, which is known to associate with contact sites, could co-operate with Drp1 and Dyn2 recruitment to cause their destabilization in ischemia. IPC attenuates destabilization by preventing mt-HK2 dissociation.

### Ischemic attenuation of mitochondrial morphology changes—Role of contact sites

Modulation of IMM morphology by ANT ligands in our LS experiments (Fig 8) are thought to be a surrogate marker of contact site lability [26]. Thus, we propose that the decreased magni-tude and slow rate of ANT ligand-induced changes in LS exhibited by mitochondria from ischemic hearts reflect lower contact site stability induced by ischemia and their maintenance by IPC. Indeed, Bakker et. al [60], through the use of stereological methods, demonstrated that the ability of heart mitochondria to form contact sites is lost during ischemia, worsening the injury at reperfusion [60]. Our own data showing reduced rates of the creatine phosphate shut-tle during reperfusion after prolonged ischemia and the prevention of this effect by IPC are consistent with this proposal [13]. Furthermore, using permeabilized cardiomyocytes we were able to show that mt-HK2 dissociation from mitochondria using G6P and low pH attenuated creatine shuttle activity implying that stabilization of contact sites with mt-HK2 is important for the efficient function of the shuttle [13]. However, removal of mt-HK2 had no significant effect on the $Ca^{2+}$ and ANT ligand-induced effects on LS of isolated mitochondria (Fig 8), implying that these morphological changes are reflecting more than just contact site destabili-zation. The TEM data of Fig 6 confirm this and show that LS changes are accompanied by changes in the cristae structure and IMM morphology. IMM remodeling is required during fission to ensure the proper segregation of IMM at both ends of the Drp1 constriction stalk before a fission event. It is possible that the reduced LS responses seen in mitochondria at the end of ischemia may indicate that, despite binding of Drp1 (Fig 5), the mitochondria have undergone sub-optimal fission [61]. This process is known to lead to cytochrome c loss and so IPC, by maintaining mitochondrial membrane contact sites may ensure IMM remodeling is complete, thus preventing loss of cytochrome c.

### Limitations of the current study and future avenues

Although the approach chosen to dissociate mt-HK2 in the current investigations can be con-sidered physiological-like (low pH and high levels of G6P) it was performed *ex vivo* and out-side the cellular context. While this gave us control over the dissociation process it does not

allow exploration of any potential mechanistic link between mt-HK2 release and Dyn2 translocation. However, the role and regulation of fusion / fission proteins in mPTP opening is complex and controversial and to investigate this interaction more fully is beyond the scope of our present study. We provide data that are suggestive of an involvement of Drp1 and Dnm2 which we hope may be a stimulus for further studies such as investigating the signaling pathway(s) that modulate mt-HK2 binding and Dyn2 translocation to mitochondria during ischemia and IPC. It might also be illuminating to employ genetic models of altered HK/HK2 expression, but their utility is subject to significant limitations. First, overexpression of HK2 generally increases the cytosolic pool rather than promoting mitochondrial association of HK2 [62], consequently there is an increase in glycogen content [63], which is considered a cardioprotective regimen. Second, knock-out could produce profound metabolic effects that might mask the effects of acute mt-HK2 release on mitochondrial morphology / function. In this regard, there is currently a need for the development of more efficient approaches to promote acute mt-HK2 release as the available drug- and peptide-based procedures have significant side-effects [15, 17, 18, 64].

Although we observed a concomitant association of Dyn2 with mitochondria when mt-HK2 dissociates during ischemia, and that this effect is prevented by IPC, we cannot rule out the possibility that the effect is independent of mt-HK2 release and represents a parallel effect rather than a downstream event. Similarly, it would be of major importance to evaluate in greater detail how Dyn2 and/or Drp1 modulate mitochondrial ultrastructure which was presently only assessed using light scattering.

## Conclusion

We have shown that release of mitochondrial-bound HK2 *in vivo* during ischemia does not affect end-ischemia mitochondrial function directly. Rather, we propose the effects are indirect through destabilization of contact sites between mitochondrial inner and outer membranes. These are thought to be the site of HK2 binding and are important for the efficient operation of the creatine shuttle. IPC, by maintaining mt-HK2 binding, stabilizes these contact sites preventing their disruption during ischemia. Ischemia also causes the recruitment of Drp1 and Dyn2 to mitochondria that together mediate changes in mitochondrial morphology and cristae structure to enhance cytochrome c release and sensitizes mPTP opening on reperfusion. We propose that mt-HK2 binding to contact sites may prevent Drp1-mediated recruitment of Dyn2 thus explaining how IPC exerts its cardioprotective effects.

## Supporting information

**S1 Fig. Diagram depicting the experimental design and protocol for mitochondrial preparation and treatment.** Note that the dissociation protocol used to release mitochondrial-bound HK2 occurs prior to data acquisition and that dissociation agents are absent from the washing buffers.
(TIFF)

**S2 Fig. Effect of washing buffer pH on mt-HK2 dissociation assessed through the activity of the remaining mt-HK.** Total HK activity of mitochondria pre-treated with dissociation buffer during isolation protocol (a); data are presented as mean ± SEM of 4 independent experiments on different mitochondrial preparations. Differences between groups were evaluated by a matched pair (± low pH) t Student's test. (b) uncropped and representative immunoblot against HK2 on isolated mitochondrial fractions. (c) Coomassie staining of membrane shown in (b). The loading control for membrane in (b) is shown on S6b Fig (the mitochondrial

protein ANT). (d) uncropped immunoblots against HK2, HK1 and ANT as shown on main Fig 1.
(TIFF)

**S3 Fig. Protective effects of 0.2 μM cyclosporine A (CsA) on the sensitivity of mPTP opening under de-energized conditions.** Mitochondrial swelling associated with mPTP opening was triggered by addition of 83.5 μM free $Ca^{2+}$ and monitored by measurement of $A_{520}$ (a). Data are presented either as mean of 5–6 independent experiments ($b_i$-$c_i$) or overlaid with individual runs ($b_{ii}$-$c_{ii}$) for the perfusion groups in the study (pre-ischaemia—bi, bii; end-ischaemia—ci, cii; and, IPC end-ischaemia—di, dii).
(TIFF)

**S4 Fig. Protective effects of 0.2 μM CsA on the sensitivity of mPTP opening under energized conditions.** mPTP opening was evaluated by the calcium retention capacity assay with addition of 20 μM free $Ca^{2+}$ every 2 min (a). Data are presented as mean independent experiments ($b_i$-$c_i$) or overlaid with individual runs ($b_{ii}$-$c_{ii}$) for the perfusion groups in the study (pre-ischaemia—bi, bii; end-ischaemia—ci, cii; and, IPC end-ischaemia—di, dii).
(TIFF)

**S5 Fig. Representative western blots for Fig 5.** Dyn-2 (a, b), Drp1 (a, b) and Mff (c) data shown on Fig 5. Total extracts are shown in (a) while mitochondrial fractions are shown in (b), for Dyn-2 and Drp1. For Mff only mitochondrial fractions were analyzed. GAPDH was used as loading control for the total extracts while VDAC was used for mitochondrial fractions instead. The vertical white bar along the blots on (a) and (b) indicate that although the samples were in the same membrane they were not next to each other (non-relevant samples are omitted in the picture).
(TIFF)

**S6 Fig. Effect of ischemia and "low pH + G6P" on the levels of mitochondrial ANT protein.** Data are presented as mean ± SEM of 6–8 independent mitochondrial fraction samples (a). For details on the use of "low pH + G6P" please see the Material and methods section. Differences between groups were evaluated by a matched pair (± G6P low pH) two-way ANOVA with interaction followed by Holm-Šídák pos-hoc test to correct for multiple comparisons. (b) representative western blot of the analysis in (a). (c) loading control of the western blot shown on (b) by Coomassie staining.
(TIFF)

**S7 Fig. EM segmentation pipeline.** Example of the image-processing pipeline for mitochondrial membranes segmentation in EM micrographs. A—original; B—band pass filter; C—Otsu threshold; D—particle size filtering.
(TIF)

**S1 Table. Hemodynamic data monitored before ischemia of hearts used for mitochondrial isolation.** Hearts were perfused according to the protocols described in the Material and Methods section. All the data presented in the table correspond to hemodynamic function recorded prior to the index ischemia at the end of the stabilization period. Data for each parameter were analyzed by a one-way ANOVA followed by Holm-Šídák pos-hoc test to correct for multiple comparisons. *, p<0.05 vs pre-ischemia or ischemia. Abbreviations: EDP—end-diastolic pressure; IPC—ischemic preconditioning; Isch.—ischemia; RRP; rate pressure product; SP—systolic pressure.
(DOCX)

**S2 Table. Parameters relating to ischemic contracture of hearts used for mitochondrial isolation.** All the data presented in the table were obtained during the index ischemia. T0 indicates time at which rigor started; Amax, rigor maximum amplitude. Data for each parameter was analyzed by a two-tail Student's t test. *, p<0.05. Abbreviations: IPC—ischemic preconditioning; n.a.—not applicable.
(DOCX)

**S3 Table. Hemodynamic data monitored during reperfusion for the group of hearts used to study infarct size.** Hearts were perfused according to the protocols described in the Material and Methods section. All the data presented in the table correspond to hemodynamic function recorded after 5 and 60 min of reperfusion. Infarct size was assessed by the TTC staining 120min after starting reperfusion. Data for each parameter were analyzed by a two-tail Student's t test. *, $p < 0.05$; ***, $p < 0.001$. Abbreviations: AAR—area at risk; IPC—ischemic preconditioning.
(DOCX)

**S4 Table. Morphometric data from EM micrographs of isolated mitochondria incubated with different ANT ligands.** Data are presented as mean ± SD and differences relatively to baseline were analyzed by a one-way ANOVA followed by Dunnet's pos-hoc test to correct for multiple comparisons. ***, p<0.001 vs baseline.
(DOCX)

## Author Contributions

**Conceptualization:** Gonçalo C. Pereira, Andrew P. Halestrap.

**Data curation:** Gonçalo C. Pereira, Laura Lee, Nadiia Rawlings, Joke Ouwendijk.

**Formal analysis:** Gonçalo C. Pereira, Laura Lee, Nadiia Rawlings.

**Funding acquisition:** Jeremy M. Henley, Andrew P. Halestrap.

**Investigation:** Gonçalo C. Pereira.

**Methodology:** Gonçalo C. Pereira, Laura Lee, Nadiia Rawlings, Joke Ouwendijk, Joanne E. Parker, Tatyana N. Andrienko.

**Project administration:** Jeremy M. Henley, Andrew P. Halestrap.

**Resources:** Jeremy M. Henley, Andrew P. Halestrap.

**Supervision:** Jeremy M. Henley, Andrew P. Halestrap.

**Writing – original draft:** Gonçalo C. Pereira, Andrew P. Halestrap.

**Writing – review & editing:** Gonçalo C. Pereira, Andrew P. Halestrap.

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
