## [Decision Letter · Decision Letter 0]

21 Nov 2019

PONE-D-19-26403

HEXOKINASE II DISSOCIATION ALONE CANNOT ACCOUNT FOR CHANGES IN HEART MITOCHONDRIAL FUNCTION, MORPHOLOGY AND SENSITIVITY TO PERMEABILITY TRANSITION PORE OPENING FOLLOWING ISCHEMIA

PLOS ONE

Dear Dr. Pereira,

Thank you for submitting your manuscript to PLOS ONE. After careful consideration, we feel that it has merit but does not fully meet PLOS ONE’s publication criteria as it currently stands. Therefore, we invite you to submit a revised version of the manuscript that addresses the points raised during the review process.  Multiple experts in the field have reviewed the manuscript. Their comments are generally positive, and a revision that addresses the comments of each reviewer would be appreciated. Please carefully consider and discuss the comments of Reviewer 1. All reviewers requested more original Western blotting data.  Thank you for your patience and submission to PLoS One.

We would appreciate receiving your revised manuscript by Jan 05 2020 11:59PM. To enhance the reproducibility of your results, we recommend that if applicable you deposit your laboratory protocols in protocols.io, where a protocol can be assigned its own identifier (DOI) such that it can be cited independently in the future. For instructions see: http://journals.plos.org/plosone/s/submission-guidelines#loc-laboratory-protocols

We look forward to receiving your revised manuscript.

Kind regards,

Edward J. Lesnefsky, MD

Academic Editor

PLOS ONE

Journal Requirements:

"This work was supported by

 RG/08/001/24717 and PG/12/40/29634 to APH and PG/14/60/3014 to JH from the British Heart Foundation.

The funders had no role in study design, data collection and analysis, decision to publish, or preparation of the manuscript.".

i) Please provide an amended statement that declares *all* the funding or sources of support (whether external or internal to your organization) received during this study, as detailed online in our guide for authors at http://journals.plos.org/plosone/s/submit-now.  Please also include the statement “There was no additional external funding received for this study.” in your updated Funding Statement.

ii) Please include your amended Funding Statement within your cover letter. We will change the online submission form on your behalf.

Reviewers' comments:

Reviewer's Responses to Questions

**Comments to the Author**

1. Is the manuscript technically sound, and do the data support the conclusions?

Reviewer #1: No

Reviewer #2: Yes

Reviewer #3: Partly

Reviewer #4: Yes

Reviewer #5: Yes

2. Has the statistical analysis been performed appropriately and rigorously? 

Reviewer #1: Yes

Reviewer #2: Yes

Reviewer #3: Yes

Reviewer #4: Yes

Reviewer #5: Yes

3. Have the authors made all data underlying the findings in their manuscript fully available?

Reviewer #1: No

Reviewer #2: Yes

Reviewer #3: Yes

Reviewer #4: Yes

Reviewer #5: Yes

4. Is the manuscript presented in an intelligible fashion and written in standard English?

Reviewer #1: Yes

Reviewer #2: Yes

Reviewer #3: Yes

Reviewer #4: Yes

Reviewer #5: Yes

5. Review Comments to the Author

Reviewer #1: In this report, the authors study the role of in vitro HK2 dissociation in mitochondrial functions after ischemia. The authors conclude that in vitro HK2 dissociation alone does not replicate ischemia induced effects of mitochondrial function and morphology. The experiments appear to be carefully performed and data presentation is extensive. The paper, however, presents little that is unexpected, is largely descriptive and correlative, and leaves gaps that limit the impact and credibility of the conclusions. The study unfortunately does not examine this signaling axis in an in vivo system and thus the relevance of the finding is of minimal impact. There are also shortcomings in the experimental design.

1. It has been established by this lab and others that mt-HK2 provides protection against ischemic stress (eg., PMID 21527739, 23836898, 21071708) and, in general, it is not surprising that inhibition of protective pathway is not sufficient to induce pathophysiological responses. Indeed, HK2 KO mice (heterozygous) are viable, fertile, and grew normally (PMID 10428828). It has also been demonstrated that at baseline, HK2+/− mice have normal cardiac function but display lower systolic function after I/R (PMID 21071708). Thus novelty regarding mt-HK2 is somewhat incremental.

2. This paper uses low pH and G6P to dissociate HK2 from isolated mitochondria, solely relying on this intervention. The specificity of the treatment is not demonstrated although the treatment would have other effects on isolated mitochondria. More specific interventions should also be utilized. For instance mt-HK2 dissociation peptides have been widely used by many investigators in cells, ex vivo heart and in vivo heart (PMID 15574336, 21527739, 31570704) and this should be more selective, especially in isolated mitochondria.

3. A number of relevant work studying the effect of mt-HK2 in the heart (for examples, some of papers cited in the above) are not even mentioned in this manuscript. The findings of these papers as well as the contradictions they might generate should be adequately discussed. In addition, there are conflicting reports regarding the role of mitochondrial fusion/fission in I/R injury. For example, Ikeda et al (PMID 25332205) showed that the infarct size after I/R was significantly greater in cardiac-specific Drp1 heterozygous knockout than in control mice. This study also showed that mdivi-1 has direct cell-protective effects on cardiomyocytes independent of Drp1. Studies using Mfn-2 knockout suggest that Mfn-2, a fusion protein, serves to predispose cells to mitochondrial permeability transition and to trigger cell death (PMID 21245373, 22037195).

The authors need to describe about these issues in a balanced fashion.

4. The effect of mt-HK2 dissociation is examined only in isolated mitochondria in this study. Although this allows precise examination of the effect of mt-HK2 dissociation on mitochondrial function, it would not mimic the environment in vivo. Thus in vivo analysis would be required to strengthen this paper.

Furthermore, what condition shown here with low pH and G6P in isolated mitochondria from ischemic heart can mimic the stage of ischemic heart diseases? G6P accumulation and lower pH occur during ischemia while calcium overloading and mPTP pore opening (figures 3 and 4) occur upon reperfusion. mt-HK2 is already decreased by ischemia through G6P accumulation and lower pH, and is further decrease in isolated mitochondria with low pH and G6P relevant as a model for vivo I/R? Does reperfusion further decrease the level of mt-HK2? If so what is the mechanism?

5. Analysis of mitochondrial morphology is also carried out only in isolated mitochondria. It is not clear what is expected in the experiment without other cellular compartments, especially cytosol. For instance, Drp1 is largely cytosolic (97%) and translocates to mitochondria by post-transcriptional modifications, and this aspect is lost in isolated mitochondria experiments. Another reason why mitochondrial morphology should also be examined in the heart (not isolated mitochondria), for example, by electron microscopic analysis in heart sections, is that conventional mitochondrial fractionation might not efficiently collect fragmented- and small mitochondria (PMID 25600785).

6. Other mitochondrial fission proteins (Fis1, Mid49 and Mid51) should also be examined. How about mitochondrial fission proteins (Mfn1/2, Opa1)?

7. VDAC is used to normalize Mff, Drp1 and Dyn2 levels in the mitochondrial fraction (Figure 5) . If OMMP is induced by ischemic stress, inner mitochondrial protein would be more appropriate as a control.

8. Representative Western blots should be shown.

Reviewer #2: Here, the authors test whether the loss of HK2 from mitochondria is sufficient for the effects associated with this process on MPTP. Recapitulation of the enhanced G6P/reduced pH conditions associated with ischemia dissociated the vast majority of HK2 from mitochondria isolated from pre- and -end ischemic hearts as well as PCed hearts. However, this did not alter respiration, OMMP, or mPTP opening in any of the groups, which still exhibited the expected changes between pre, end and PC hearts. Mitochondrial translocation of Drp1 and increased mitochondrial size was observed in the end-ischemic group revealed translocation to mitochondria during ischemia but was unaffected by PC or loss of HK2. The authors conclude that loss of HK2 during ischemia is not responsible for all the observed mitochondrial effects, and that other mechanisms are at play.

This is a thorough, well designed study that uses multiple, overlapping indices to truly address an important question that has remined in the field for quite some time.

The study is well controlled, sufficiently powered, and the statistical analyses are appropriate and rigorous

Comments

1) Protein expression of Drp1 is not necessarily indicate of activity. Assessment of Drp1 Ser616 phosphorylation would provide more insight in this regard. For example, PC may not alter translocation, but it might alter phosphorylation.

2) The Westerns should be included with the quantified data in the main figures. It would also be helpful if the dilution used could be included in the methods.

Reviewer #3: In this paper, Pereira et al. studied whether HK2 release from the mitochondria is the primary signal mediating ischemia-induced mitochondrial dysfunction. They isolated mitochondria from Langendorff-perfused rat hearts before and after 30 min of ischemia ± ischemic preconditioning (IPC). They then subjected the isolated mitochondrial to in vitro dissociation of HK2 by incubation with glucose-6-phosphate at pH 6.3. HK2 dissociation had no effect on their cyt-c release, respiration or mPTP opening. They also noted no major ultrastructural differences between pre- and end-ischemia mitochondria, but the amplitude of changes in light scattering was reduced in the mitochondria after end-ischemia. They also showed more Drp1 in end-ischemia mitochondria, but IPC failed to prevent this increase but did decrease mitochondrial-associated dynamin 2. They concluded: “In vitro HK2 dissociation alone cannot replicate ischemia-induced effects on mitochondrial function implying that in vivo dissociation of HK2 modulates end-ischemia mitochondrial function indirectly perhaps involving interaction with mitochondrial fission proteins. The resulting changes in mitochondrial morphology and cristae structure would destabilize outer / inner membrane interactions, increase cyt-c release and enhance mPTP sensitivity to [Ca2+].”

These studies are interesting and will advance the field. However, the authors solely relied on G6P and pH of 6.3 to dislocate HK2 from the mitochondria. These processes have other effects, including inhibition of hexokinases. The authors should have used better methods to dislocate HK2 form the mitochondria, including the use of clotrimazole or a peptide specific to the mitochondrial binding domain.

Additionally, the actual western blots measuring protein levels are missing in some figures.

Reviewer #4: Summary: The investigators showed previously that cardiac ischemia results in HK2 dissociation from mitochondria with concomitant loss of cyt c and induction of mPTP opening on reperfusion. The loss of HK2 is thought to be due to the increase in glu-6-P and decrease in pH during ischemia caused by increased glycolysis. IPC is somewhat protective with the hypothesis that IPC might reduce the dissociation of HK2 from mitochondria due to less glycolysis during ischemia with subsequent improved cardiac function on reperfusion and a decrease in infarct size. Cyt c release and the juxtaposition of OMM and IMM and their components VDAC and ANT, and of HK2 with VDAC, figure into sensitization of mPTP opening with IR injury. Using an isolated mitochondrial model, the authors tested the hypothesis that the loss of HK2 attachment to mitochondria is the major factor in predisposing to cell damage because IPC reduces detachment of HK2 from mitochondria. To rule out cytosolic factors that might interact with HK2 or the OMM they examined mitochondria in isolation. In their model they isolated mitochondria from non-ischemic isolated hearts and from hearts subjected to IR injury ± IPC. In each group of hearts HK2 was artificially dissociated from mitochondria using an acidic buffer (pH 6..3) plus an excess of glu-6-P. They then measured common indices of mitochondrial function and integrity, such as respiration, CRC (Fura-FF), membrane potential, and OMM and cristae morphology (by light scattering-swelling assay and TEM) with manipulation of ANT conformations using CAT. In other experiments they did not artificially dissociate HK2 from mitochondria, but used western blotting after IR± IPC to assess changes in mitochondrial amounts of HK2, ANT, Dyn2 and Drp1., using antibodies against these proteins, and measured activities of HK and citrate synthase.

They found that THE chemically induced dissociation of HK2 did not mimic the mitochondrial changes found after IR injury as there was no effect on cyt c release, respiration, or proneness to mPTP opening. So, alternatively, they suggested that there is an intermediate step during IR injury that leads to HK2 dissociating from mitochondria. They furnish evidence that IR modulates the mitochondrial binding of two other proteins, Drp1 and Dyn2, based on their finding that Drp1 increased in mitochondria after ischemia (not affected by IPC) but that an increase in Dyn2 was reduced by IPC. From this they speculated that Drp1 and Dyn2, interacting with HK2 on the OMM, cause morphological changes that destabilize the OMM to enhance cyt c release and mPTP sensitivity to elevated (matrix) calcium. The authors speculate that Drp1 translocation from the cytosol to mitochondria occurs during ischemia and this in turn recruits Dyn2 to the OMM where it is tethered if HK2 is no longer bound. The binding of Dyn2 in the absence of HK2 is proposed to destabilize the OMM and IMM. The authors conclude that normally HK2 binding to contact sites on the OMM may prevent Drp1 from recruiting Dyn2 and that IPC is cardioprotective because it prevents the increase in Dyn2 in mitochondria. They concede that HK2 dissociation and association of Dyn2 may be unrelated, but parallel events in IR injury but posit that Drp1 and Dyn2 cannot associate with the OMM if HK2 is present.

I have only a few comments:

1. The manuscript is well written with good description of all the methods, results and statistics. The discussion is balanced and admits that the connection between HK2 dissociation and Dyn2 association is unclear and needs to be better discerned. The plan to isolate the role of HK2 dissociation in IR injury by simulating HK2 dissociation in isolated mitochondria devoid of cytosolic factors after IR injury was good.

2. How do you explain the almost complete loss of HK2 content with only a 50% reduction of HK activity? Is there HK1 activity associated with bound HK1?

3. Fig. S4 should be placed in the main text as it is pertinent to the role of this protein.

4. The investigators did an extraordinary amount of work with the LS technique and TEM to assess the morphological changes (Figs. 6-8) after ANT ligand treatment and artificially induced HK2 dissociation after ± IR ± IPC. The problem is I don’t clearly understand the connection of this data with the morphological effects of HK2 dissociation, or not, and the roles of Drp1 and Dyn2. Can you please clarify this better for the reader? I’m referring to lines 557-564.

5. What is the Drp1 or Dyn2 or Mff distribution pattern in the pre ischemia, end ischemia and IPC, end ischemia in the presence and absence (acidic pH +glu-6-P) of HK2?

6. Did you correlate LS data and IMM morphology data (TEM data) during IPC? This could have given some insight into the effect of IPC on the preservation of mitochondrial cristae dynamics (if any?). In addition, do you have results on the presence and absence (acidic pH + G6P) of HK2 during IPC? This could furnish information on the effect of HK2-dependent Drp1/ Dyn2 recruitment to mitochondria.

7. Fig. 8b and 8d: Shouldn’t these numbers be negative?

Reviewer #5: Hexokinase II (HKII) catalyzes the conversion of glucose to glucose-6-phosphate (G6P). Studies have shown that HKII inhibits cell death including during myocardial ischemia-reperfusion injury. During ischemia, mitochondrial HKII dissociates from mitochondria, and the extent of mitochondrial HKII dissociation is correlated with cell death. However, it remains unclear whether the dissociation of HKII from mitochondria is correlative or causative with respect to cardiac damage during ischemia-reperfusion. Using isolated mitochondria from rat hearts subjected to ischemia ex vivo, Pereira et al. report that HKII dissociation alone is not sufficient to account for morphological and functional abnormalities following ischemia.

Comments:

1. Fig. 1. First, the HKII Western data in Supp Fig 1 should be included in Fig. 1. Second, is the persisting HK activity in low pH + G6P specific to HKII? If it is, this would suggest that not all the HKII has been dissociated? Is there a specific HKII inhibitor that could be used to show this residual is not attributable to remaining HKII? Third, if this residual HK activity is real and attributable to HKI (as you posit), is HKI also known to inhibit cell death? If so, levels of HKI should also be assessed by Western. The reason for asking these questions is that the rest of the study shows largely negative data. In order to interpret this negative data definitively, it is necessary to know that low pH + G6P is, in fact, inactivating protective mechanisms resulting from all HKs at the mitochondria.

2. Fig. 3b. Please check the comparison in the ischemia group (red bars) between normal and low pH + G6P (all no CsA). Is this really 3 stars?

3. Fig. 5. I do not understand how these data showing how ischemia or ischemic preconditioning affects the abundance of various mitochondrial fission components at the mitochondria relate to the rest of the story, which is about HKII dissociation from mitochondria not playing a causal role in cell death. These data are not really developed. Unless I missed your point, why include this figure in the paper?

4. Figs 6 and 7. This is validation of an approach. It is important, but could be placed in supplemental data. The actual data using this approach, shown in Fig. 8, is what belongs in the main paper.

5. Fig. 8. I understand your main point that the normal versus low pH + G6P pairs showed no difference. But, why was delta-Abs520 under Ca2+ treatment conditions not rescued by IPC compared with ischemia alone?

6. PLOS authors have the option to publish the peer review history of their article (what does this mean?). If published, this will include your full peer review and any attached files.

Reviewer #1: No

Reviewer #2: No

Reviewer #3: No

Reviewer #4: Yes: David F. Stowe, MD, PhD and Jyotsna Mishra, PhD

Reviewer #5: No

---

## [Author Response · Author response to Decision Letter 0]

15 Dec 2019

Reviewer #1

General Comments 

We are pleased to see that the Reviewer, although critical, recognizes that the experiments have been “carefully performed and data presentation is extensive”. We were unsure why the reviewer feels that the paper presents little that is unexpected. A key message of the paper is that despite the strong correlation between mt-HK2 binding at the end of ischemia and cardioprotection, this effect cannot be a direct effect of HK2 on the mitochondrial permeability transition and must be through downstream mechanisms. We think it is a little strong to say this was expected since there had been extensive literature on the role of HK2 on mPTP function in the literature. It is also a little harsh to say the data are largely descriptive; there are extensive quantitative data to back up the conclusion that HK2 itself is not the key regulator of mPTP opening in ischemia / reperfusion. We also take issue with the criticism that a major shortcoming is the signalling axis in vivo. We previously addressed this and showed that the extent of HK2 dissociation during ischemia could be explained by the magnitude of the drop in pH and build up of glucose-6-P during ischemia. The purpose of this paper was to establish how the change in HK2 binding related to the sensitivity of the mPTP to opening at the end of ischemia and we believe we do this convincingly.

Specific Comments

1. We would agree with the reviewer that there is much published data to support the cardioprotective role of mtHK2 and it was our own published work (PMID 23329796, 23525412, 25204670) on this that has led to the present studies. However, what these earlier studies do not address is whether the protective effects of mtHK2 are a result of direct inhibition of mPTP opening on reperfusion. Here we show that this cannot be the case and we would suggest that this is a very significant observation that is more than incremental. Moreover, genetic ablation rather than acute mt-HK2 dissociation will probably induce metabolic remodelling in the cardiac tissue that is secondary to HK2 redistribution in the cell, masking the real effect of acute mt-HK2 dissociation.

2. The reviewer is critical of our use of low pH and G-6-P to dissociate mtHK2 and suggest the use of HK2 dissociating peptides. However, our previous studies have shown that the use of such peptides is open to criticism, especially in vivo, because of non-specific effects (PMID 23329796). More importantly, we have previously shown that the mechanism of HK2 dissociation during ischemia involves the observed decrease in pH and increase in [G-6-P], both of which are attenuated by ischemic preconditioning. Thus, we would argue that the use of low pH and G-6-P to dissociate HK2 in isolated mitochondria is entirely appropriate.

Additionally, we are not sure what the reviewer meant by “the treatment would have other effects on isolated mitochondria”. The combination of low pH and G-6-P is: 1) a physiological phenomenon; 2) we have previously demonstrated (PMID: 23525412), and confirm here, that none of the effects alone can affect mitochondrial respiration; 3) the dissociation protocol occurs prior to assaying mitochondrial function, decreasing any interference during data acquisition; 4) the dissociation protocol is brief and occurs at low temperature having negligible effects on mitochondrial metabolism. 

We also think that the use of a peptide-based approach is not straight forward. For example: solubility of peptides in aqueous solutions can be challenging, lowering their effective concentration; 2) reaction or sample vessels require coating to prevent the peptides to bind to their surface; 3) peptides and peptides-based approach are prone to proteolysis both by internal and external proteases present during mitochondrial isolation. 

Overall, we agree with the reviewer that peptide-based approaches are certainly useful in several biological setups; however, we are confident that our approach is as valid if not better suited for the present investigations.

3. The reviewer suggests that we cite more earlier papers on mtHK2 and discuss their data in relation to our present studies. We have previously discussed this work in our earlier papers cited here (PMID 23329796, 23525412, 25204670) but now also refer to data implicating HK2 in the regulation of mPTP opening in other systems. Please see lines 93-95 in the revised manuscript. We suggest that a wider ranging discussion on the possible role of the fission fusion machinery in regulating the mPTP is outside the scope of the present paper but in line 718 we now refer to an earlier review in which we do this (PMID 25204670). Furthermore, in response to Reviewer #2 we do now discuss the possible role of Drp1 phosphorylation. This is provided in lines 726-733 of the revised manuscript.

4. The reviewer suggests that in vivo analysis of mtHK2 dissociation would strengthen the paper, but we are not sure in what way. We and others have already provided evidence that the extent of mtHK2 dissociation in vivo at the end of ischemia correlates with the damage on reperfusion and on the sensitivity of mitochondria isolated at the end of ischemia to mPTP opening and cytochrome c release. The purpose of this paper is to establish whether the dissociation of mtHK2 alone is what mediates these effects. In our opinion, this is best done with isolated mitochondria where interference by other outer membrane and mPTP interacting factors might complicate data interpretation.

The other point raised by the reviewer relates to the fact that we are measuring changes occurring in ischemia whereas the damage through mPTP opening occurs on reperfusion. This is a critical point. We showed previously (PMID 18356542) that it is the mitochondrial state at the end of ischemia that determines their sensitivity to mPTP opening on reperfusion. Thus, mitochondria isolated from ischemic hearts are much more sensitive to mPTP opening than those from control hearts. However, ischemic or pharmacological preconditioning attenuates this sensitisation (PMID 18356542, 20558443) and thus leads to less mPTP opening on reperfusion and consequently less injury. More recently (PMID 27907091) we showed that ischemic preconditioning did not attenuate the elevated [Ca2+] at the end of 30 min ischemia or ROS production in the first 90s of reperfusion yet mPTP opening after 1 min and subsequent ROS production was attenuated. This confirms that it is something happening to mitochondria during the ischemia that determines mPTP opening and thus the extent of injury on reperfusion and HK2 binding was the only parameter that we measured that correlated with this. It is for this reason that we focussed our attention on end ischemic mitochondria. Measurements during reperfusion would be hard to interpret because during this phase mPTP opening occurs with secondary changes in many other parameters including pH, ROS and [Ca2+]. The fact that mt-HK2 from end-ischemic hearts can be further reduced in vitro indicates that the ischemic insult was not extreme, suggesting that further damage could have been attained to the heart. This agrees with our previous findings (PMID: 23525412) where variations to the standard ischaemic protocol led to different amounts of mt-HK2 being retained at end of ischaemia, correlating with cardiac damage at reperfusion.

5. We appreciate the reviewer’s comments on the desirability of investigating mitochondrial morphology in situ. We initiated such studies but soon recognised that the major technical and analytical challenges of such an approach were beyond the scope of the present study. However, we would point out that our TEM and light scattering studies on isolated mitochondria directly correlate with the other studies we make on isolated mitochondria and thus do have value in their own right. We must leave it to others with greater expertise to explore mitochondrial morphology in situ.

6. With regards to Drp1, it is its translocation to mitochondria that induces fission and thus its presence in isolated mitochondria rather than the cytosol that is relevant. 

We agree that working with isolated mitochondria has its pitfalls, but we are aware of those limitations and take them into account during the interpretations of our experimental data. With regards to the particular point raised by the reviewer, that conventional mitochondrial fractionation might not efficiently collect fragmented- and small mitochondria, we would note the following. 1) our TEM morphometric analysis shows that the size of our mitochondrial population spans 1-2 orders of magnitude; 2) we use a protease-based extraction in combination with differential centrifugations to collect both sub-sarcolemma and interfibrillar mitochondria; 3) the protocol we use, contrary to that used by others including the one on the reference provided by the reviewer, include a Percoll gradient to significantly reduce cellular contaminants such as endomembrane systems, broken membranes and, importantly, smaller vesicles with different buoyant densities compared to mitochondria.

7. We agree that other fusion / fission factors could be studied but we focussed on those that had already been implicated in ischemia reperfusion injury and so might provide a possible mechanism for regulating mPTP opening via mitochondrial morphology. An extensive study of this is beyond the scope of the present study but we hope that our data may inspire others to perform more detailed studies.

8. The reviewer suggests that an inner membrane marker might be more appropriate than VDAC. However, we are unsure why that would be the case since Mff, Drp1 etc are associated with the outer membrane which is the location of VDAC. Furthermore, permeabilisation of the outer membrane would not lead to loss of VDAC, but only of intermembrane proteins such as cytochrome c due to formation of membrane pores rather than breakage and loss of OMM fragments.

9. Representative Western blots were shown in the supplemental material. This information was clearly stated in Fig. 1 legend but was missing in that of Fig. 5. This has been added in the updated version of the manuscript.

Reviewer#2

General Comments 

We thank the Reviewer for the positive assessment of the paper.

Specific comments

1. We appreciate the reviewer’s comment about Drp1 phosphorylation, and in retrospect this would have been a good additional measurement. Unfortunately, we no longer have the samples to perform the analysis but do now discuss this possibility in lines 727 of the revised manuscript. However, we note that for preconditioning to exert an effect via phosphorylation of Ser616, which enhances fission, it would have to cause dephosphorylation. We have previously performed extensive studies of protein phosphorylation in mitochondria isolated from control and preconditioned hearts, pre-ischemic, end ischemic and reperfused, and found no evidence of a phosphorylated protein present at the expected MWt for Drp1 (see PMID 18356542). We also feel that the literature itself is not that clear about the effects of different phosphorylations of Drp1 as we now note in our discussion.

2. Cropped versions of Western Blot data have been added to the main figures to show representative blots. The full non-cropped membranes are shown in the supplemental material as requested by PlosOne guidelines.

Reviewer#3

General Comments 

We thank the Reviewer for their positive assessment of the paper.

Specific comments

1. The reviewer is critical that we only use low pH and G-6-P to dissociate mtHK2 and suggest the additional use of clotrimazole and HK2 dissociating peptides. However, our previous studies have shown that the use of such peptides is open to criticism because of non-specific effects (PMID 23329796). More importantly, we showed that the mechanism of HK2 dissociation during ischemia involves the observed decrease in pH and increase in [G-6-P], both of which are attenuated by ischemic preconditioning. Thus, we would argue that the use of low pH and G-6-P to dissociate HK2 in isolated mitochondria is the most appropriate technique to use (please see answer to Reviewer#1 Comment#2 for more details).

The reviewer also points out the inhibitory effects of low pH and G-6-P on hexokinases activity. In this regard we would like to stress the following: 1) the HK2 dissociation protocol occurs prior to data acquisition and therefore, even if the inhibition and release were two possible events they do not play a role during data acquisition where conditions were identical for all treatments; 2) none of the buffers used in the present investigations contain glucose. In conclusion, when taken together with point 1, we strongly believe that hexokinase(s) activity has no effect on the interpretation of our data. These assumptions would be more difficult to prove if chemical drugs were used, because they might well induce off target effects.

2. Cropped versions of Western Blot data have been added to the main figures to show representative blots. The full non-cropped membranes are shown in the supplemental material as requested by PlosOne guidelines.

Reviewer#4

General Comments 

We thank the Reviewer for their clear summary of our paper and their positive assessment of our data.

Specific comments

1. We appreciate that the Reviewer regards the paper as well presented and our approach appropriate.

2. The Reviewer is correct; HK1 is maintained when HK2 is dissociated by low pH and high [G-6-P] as occurs with ischemia in vivo. We have demonstrated this previously (PMID 21410437 23525412) and now make this clear on line 402-404 and by including a blot for HK1 on Fig. 1. Representative blots of experiments where mitochondria were treated with low pH ± G-6-P are now also included on Fig. 1 (full blots in the supplemental Figure S1), showing negligible effect on its release following the treatment. However, because data for this immunoblot was not collected as extensively as the others in the study, it is not possible to perform any statistical analysis regarding HK1 protein levels.

3. Cropped versions of Western Blot data (Fig. S4) have been added to the main figures to show representative blots. The full non-cropped membranes are shown in the supplemental material as requested by PlosOne guidelines.

4. We thank the Reviewer for highlighting our lack of clarity here and in the revised manuscript (lines 550-555, 623-627 and 665-667) we have endeavoured to explain the proposed relationships between light scattering, morphology and HK2 dissociation more clearly.

5. We did not perform experiments investigating the effects of low pH and G-6-P on the distribution of the fission/fusion proteins. Perhaps in retrospect we should have done this but our argument was that changes in Drp1 and Dynamin 2 would only occur in the presence of cytosol from where they translocate to and from mitochondria. Since the Drp1 binds in ischemia when the pH is low and G-6-P high exposing end ischemic mitochondria to these factors would not have been expected to dissociate Drp1 and without cytosolic Drp1 additional binding would not be possible.

6. The correlation of LS and TEM was only made for control mitochondria due to the laborious and time-consuming character of the latter technique. Initially we wished to establish whether there were any significant IMM morphology differences between control and ischemic mitochondria at baseline and this was why we looked at those two conditions first. However, and to our surprise, the lack of differences in these two major groups suggested we would be unlikely to detect significant changes with IPC. Therefore, since analysing TEM data is extremely labour intensive, we focused our efforts on LS. As noted under Point 4 above we have now endeavoured to explain our rationale more clearly.

In relation to the other comments made here by the reviewer it is unclear to us why TEM of IPC mitochondria would provide insight into the “preservation of cristae dynamics” if there was no difference detected after ischemia? The TEM snapshot at baseline shows the ultrastructure of mitochondria after isolation (which may not represent the status attained in situ) while the LS data suggests that it is the transition between different conformations that is significantly depressed. The reviewer also asks if we have results in the presence and absence (acidic pH + G6P) of HK2 during IPC. We are not sure to which data the reviewer is referring. All the data we have collected are shown in the present manuscript. If the reviewer is asking for TEM of IPC mitochondria +/- low pH plus G-6-P we do not have such data. However, as noted above we believe that the LS data we provide is more informative in providing information on subtle changes in membrane morphology and dynamics. We are unclear how either technique could provide information on the effect of HK2-dependent Drp1/ Dyn2 recruitment to mitochondria. 

7. Changes in absorbance values were calculated as the absolute value (independent of sign) of the difference between start and end; however, the axis legend was misleading. We thank the reviewer to point it out and we corrected the graphs to show “Absolute Amplitude”. We also made it clear in the figure legend of Fig. 8.

Reviewer#5

General Comments 

We thank the Reviewer for their clear summary of our paper.

Specific comments

1. We now include the Western blot of Supplementary Fig. 1into the main Figure. We appreciate the point being made about the importance of confirming the specificity of HK2 dissociation and that there are no changes in HK1 Representative blots of two separate experiments where mitochondria were treated with low pH plus G-6-P are now included on the supplemental figure S1, showing negligible effect on its release following the treatment. However, because data for HK1 was not collected routinely, it is not possible to perform any statistical analysis regarding HK1 protein levels. 

2. We checked our analysis for the data shown on Fig. 3b, specifically the comparison pointed out by the reviewer. We can confirm that there was no error reporting the statistical difference. We provide below a summary of our ANOVA table as well as the summary for the multiple comparisons so the reviewer and anyone can confirm our analysis. 

Two-way RM ANOVA Matching: Across row 

Assume sphericity? Yes 

Alpha 0.05 

Source of Variation % of total variation P value P value summary Significant? 

Interaction 0.8740 0.0003 *** Yes 

Perf 79.20 <0.0001 **** Yes 

G6P 0.3044 0.0017 ** Yes 

Subject 18.92 <0.0001 **** Yes 

ANOVA table SS DF MS F (DFn, DFd) P value

Interaction 0.02233 5 0.004466 F (5, 26) = 7.021 P=0.0003

Perf 2.024 5 0.4047 F (5, 26) = 21.77 P<0.0001

G6P 0.007777 1 0.007777 F (1, 26) = 12.23 P=0.0017

Subject 0.4833 26 0.01859 F (26, 26) = 29.22 P<0.0001

Residual 0.01654 26 0.0006361 

G6P - normal Mean 1 Mean 2 Mean Diff SE of diff N1 N2 t DF Adjusted P value

Stab 0.04603 0.04825 -0.002217 0.01456 6 6 0.1522 26.00 0.9848

Stab+CsA 0.03456 0.03706 -0.002500 0.01595 5 5 0.1567 26.00 0.9848

Isch 0.4649 0.5649 -0.1000 0.01456 6 6 6.869 26.00 <0.0001

Isch+CsA 0.2556 0.2722 -0.01658 0.01595 5 5 1.039 26.00 0.8356

IPC 0.09800 0.1148 -0.01676 0.01595 5 5 1.051 26.00 0.8356

IPC+CsA 0.06716 0.06186 0.005300 0.01595 5 5 0.3323 26.00 0.9829

3. We thank the Reviewer for making us aware that we had not made the link between our HK2 data and the fission protein studies sufficiently clear and we have endeavoured to address this in the revised manuscript (lines 493-496). In outline, our previous data (PMID 23525412) had shown that the extent of HK2 dissociation from mitochondria at the end of ischemia correlated well with the sensitivity of the mitochondria to mPTP opening and cytochrome c release at the end of ischemia and to the damage to the heart on reperfusion. The question we sought to address was whether HK2 was directly responsible for these effects and by displacing HK2 in vitro we showed this was not the case. Thus, the relationship between HK2 binding and mitochondrial function must be indirect. Since Drp1 had been shown to bind to heart mitochondria during ischemia and that Drp1 has effects on mitochondrial morphology which in turn is known to influence mPTP activity, it seemed appropriate to investigate whether ischemic preconditioning, which prevents HK2 dissociation, might also prevent Drp1 and Dynamin 2 binding. We recognize that the data we provide is preliminary, but it is consistent with the hypothesis that HK2 could attenuate Drp1/Dyn2 binding to mitochondria during ischemia and so prevent the detrimental effects of ischemia on mitochondrial function. We believe that our data provide the basis for future studies to test this hypothesis and thus we would like to keep the data in the paper if possible.

4. Although we can see the point being made by the Reviewer, we would prefer to keep these data in the main body of the manuscript. We also note that none of the other 4 Reviewers asked for these data to be moved to the supplementary material.

5. The Reviewer raises an interesting point for which we cannot give a definitive answer. However, it should be noted that both ADP and carboxyatractyloside give specific and total conversion of the ANT into the ‘m’ and ‘c’ states respectively while the effects of Ca2+ may be more complex. We can speculate that a possible explanation is that both ischemic and IPC mitochondria at end of ischemia are pre-loaded with Ca2+ and therefore do not experience its effect on LS. In fact, we previously showed (PMID 27907091) that IPC did not attenuate the elevated [Ca2+] at the end of 30 min ischemia. However, since we did not perform a similar quantification in the present study, we prefer to leave such speculation unless the Reviewer wishes us to include it.

---

## [Decision Letter · Decision Letter 1]

23 Mar 2020

PONE-D-19-26403R1

HEXOKINASE II DISSOCIATION ALONE CANNOT ACCOUNT FOR CHANGES IN HEART MITOCHONDRIAL FUNCTION, MORPHOLOGY AND SENSITIVITY TO PERMEABILITY TRANSITION PORE OPENING FOLLOWING ISCHEMIA

PLOS ONE

Dear Dr. Pereira,

Thank you for submitting your manuscript to PLOS ONE. After careful consideration, we feel that it has merit but does not fully meet PLOS ONE’s publication criteria as it currently stands. Therefore, we invite you to submit a revised version of the manuscript that addresses the points raised during the review process.

Reviewers 1 and 3 continue to have substantial reservations regarding the manuscript.  I invite the authors to respond to their continued reservations.  Thank you very much for your interactions with the review process.  

We would appreciate receiving your revised manuscript by May 07 2020 11:59PM. To enhance the reproducibility of your results, we recommend that if applicable you deposit your laboratory protocols in protocols.io, where a protocol can be assigned its own identifier (DOI) such that it can be cited independently in the future. For instructions see: http://journals.plos.org/plosone/s/submission-guidelines#loc-laboratory-protocols

We look forward to receiving your revised manuscript.

Kind regards,

Edward J. Lesnefsky, MD

Academic Editor

PLOS ONE

Additional Editor Comments (if provided):

Reviewers 1 and 3 continue to have substantial reservations regarding the manuscript. I invite the authors to respond to their continued reservations. Thank you very much for your interactions with the review process.

Reviewers' comments:

Reviewer's Responses to Questions

**Comments to the Author**

1. If the authors have adequately addressed your comments raised in a previous round of review and you feel that this manuscript is now acceptable for publication, you may indicate that here to bypass the “Comments to the Author” section, enter your conflict of interest statement in the “Confidential to Editor” section, and submit your "Accept" recommendation.

Reviewer #1: (No Response)

Reviewer #2: All comments have been addressed

Reviewer #3: (No Response)

Reviewer #5: All comments have been addressed

2. Is the manuscript technically sound, and do the data support the conclusions?

Reviewer #1: No

Reviewer #2: Yes

Reviewer #3: Partly

Reviewer #5: Yes

3. Has the statistical analysis been performed appropriately and rigorously? 

Reviewer #1: Yes

Reviewer #2: Yes

Reviewer #3: Yes

Reviewer #5: Yes

4. Have the authors made all data underlying the findings in their manuscript fully available?

Reviewer #1: Yes

Reviewer #2: Yes

Reviewer #3: Yes

Reviewer #5: Yes

5. Is the manuscript presented in an intelligible fashion and written in standard English?

Reviewer #1: Yes

Reviewer #2: Yes

Reviewer #3: Yes

Reviewer #5: Yes

6. Review Comments to the Author

Reviewer #1: 1. Significance: The authors claimed that this study for the first time demonstrates that mtHK2 does not directly inhibit mPTP pore while previous studies show that mtHK-II provides mitochondrial protection against mPTP. However, similar findings have been obtained using a mtHK2 dissociation peptide, gene knock-down or knock-out in isolated mitochondria, in cells and in vivo as I suggested before. More specifically, it has been shown that decrease in mtHK2 itself does not induce adverse effects, suggesting that mtHK2 dissociation itself does not trigger mPTP. mtHK2 is a negative modulator but not a component of the mPTP thus it is not surprising that the lack of this negative regulator does not affect Ca -induced mPTP opening in isolated mitochondria. In addition, although there could be potential problems in the interventions used in previously studies as the authors claim, it is not proven that the intervention used in this study (G6P and low pH) is selective or more selective than other interventions. It would be interesting to examine whether the sensitivity of the mPTP to Ca is different in cardiomyocytes and other cells which expresses less or no HK2 (eg brain and hepatocytes) and if adding HK2 protein to isolated mitochondria change it.

2. Related to point 1, what is shown here is that the dissociation does not affect Ca-induced mPTP opening, which indicates that the decrease in mtHK2 does not facilitate Ca-induced mPTP opening. This observation does not however directly address the question whether the mPTP inhibition mediated by the increase in mtHK2 is direct effect or not, a question that the authors aim to address. In addition to loss-of-function approach, gain-of-function and add-back experiments would be required to address this question.

3. As I pointed out before, the conclusion achieved by the authors solely relies on a single intervention (G6P and low pH) in isolated mitochondria. The specificity of the intervention is not demonstrated as pointed above. Low pH has been suggested to affect the sensitivity of the mPTP and G6P, a product of HK catalytic activity, inhibits hexokinase activity. mtHK1 also negatively regulates mPTP opening and this could also be affected by G6P and low pH (even it is likely to be less extent).

The authors previously showed that the TAT-HK2 dissociation impairs vascular function in Langendorff-perfused heart, although this is controversial (PMID; 23329797). The peptide has been widely used by many researchers in wide range of fields and, nonetheless, the previous observations by the authors obtained in Langendorff heart does not necessarily indicate that the HK2 dissociation peptide (without TAT sequence) has non-specific effects in isolated cardiac mitochondria.

I agree that nothing is perfectly specific, but that’s the reason why the conclusion should be supported by multiple interventions in different preparations/systems as well as by using various approaches such as gain-of-function/add-back experiments.

4. With regards to Drp1, Drp1 translocation from cytosol to mitochondria occurs through by post-transcriptional modifications including phosphorylation thus isolated mitochondria from ischemic heart (without cytosolic Drp1) is relevant to ischemic injury but not to reperfusion induced mitochondrial fission and injury. Reperfusion activates many kinases and it is shown that reperfusion increases Drp1 at mitochondria (eg., PMID; 25332205, 24477044)

The functional and mechanistic link between HK2 and mitochondrial fission proteins including Drp1 is suggested but not demonstrated by this study, although an inverse correlation is shown. As I suggested before, it used to be believed that mitochondrial fission is deleterious and fusion is protective, but recent molecular evidence suggests that this could be incorrect (PMID 25332205 21245373, 22037195). It is critical to evaluate how the changes observed in this study functionally impact on mitochondrial function/mPTP.

Reviewer #2: (No Response)

Reviewer #3: The authors have not addressed my concerns adequately. Their argument not to use a HK peptide or clotrimazole is not convincing.

Reviewer #5: Issues adequately addressed. The study has significant limitations. But these have been appropriately acknowledged.

7. PLOS authors have the option to publish the peer review history of their article (what does this mean?). If published, this will include your full peer review and any attached files.

Reviewer #1: No

Reviewer #2: No

Reviewer #3: No

Reviewer #5: Yes: Richard N. Kitsis

---

## [Author Response · Author response to Decision Letter 1]

24 Apr 2020

General Comment:

We are pleased to see that Reviewers #2 and #5 were satisfied that in our revised manuscript we had addressed all the issues raised with our original manuscript. Reviewer #4 did not respond but appeared to assess the first version of our paper positively. Thus, it would seem that three of the five reviewers are now happy that our paper is suitable for publication and we would not disagree with the comment of Reviewer #5 that “The study has significant limitations. But these have been appropriately acknowledged”. However, Reviewer #1 continues to have several significant concerns and Reviewer #3 shares one of the concerns of Reviewer #1 that we have not satisfactorily addressed why we used glucose-6-phosphate and low pH to dissociate mitochondrial HK2 rather than a peptide or drug. However, as we argue below, we are strongly of the opinion that the use of a physiological mechanism responsible for dissociation of HK2 in ischemia is more appropriate than the use of synthetic peptides or drugs that may have potential off target effects as we will explain in detail below. We strongly believe that it would be unjustified to insist on such experiments, especially in the light of three positive reviews. 

Reviewer #1

R#1_1. Significance: The authors claimed that this study for the first time demonstrates that mtHK2 does not directly inhibit mPTP pore while previous studies show that mtHK-II provides mitochondrial protection against mPTP. However, similar findings have been obtained using a mtHK2 dissociation peptide, gene knock-down or knock-out in isolated mitochondria, in cells and in vivo as I suggested before.

A1_1 - We understand the point that the Reviewer is making, but we respectfully disagree with their conclusions. Despite the numerous reports about hexokinase(s) on cell survival, either in the context of cardiovascular injury and/or cancer, none of these papers provide definite evidence of whether or how HK2 can modulate mPTP opening directly. These reports have certainly been important in illuminating potential upstream events leading to mt-HK2 release, including Akt activation and GSK3beta (in)activation, for example. However, they simply report mt-HK2 dissociation as the last event in a cascade that leads to mPTP opening and do not provide evidence that HK2 directly regulates mPTP opening. Indeed, a detailed search and analysis of literature reporting the keywords “hexokinase” plus “mitochondrial permeability” (or mPTP) show that the majority of works simply report an inverse correlation between the mtHK2 and cell death/mPTP without providing any mechanistic/molecular insight on how it attenuates pore opening. This is also true of our previous work. We are providing an Excel summary of these papers for the Reviewer’s consideration, but we do not think that to include this level of discussion in the paper would be appropriate. However, it is in the light of this literature that we felt it was of major importance to clarify whether HK2 regulates mPTP directly, which is what we endeavour to do in the present paper in the context of the ischemic / reperfused heart. It is important to note that our present investigations do not invalidate previous reports but provide significant insights into how mt-HK2 dissociation, and any parallel/concomitant events, can enhance pore opening.

R#1_2. More specifically, it has been shown that decrease in mtHK2 itself does not induce adverse effects, suggesting that mtHK2 dissociation itself does not trigger mPTP. mtHK2 is a negative modulator but not a component of the mPTP thus it is not surprising that the lack of this negative regulator does not affect Ca-induced mPTP opening in isolated mitochondria.

A1_2 - We fully agree with the Reviewer that mt-HK2 is not a classical negative modulator of mPTP opening in that its release does not lead to mPTP opening directly under basal conditions. However, we are addressing its role under conditions of ischemia reperfusion. It would be quite possible to envisage that ischemia induces an activated state of the mPTP that was blocked by bound HK2 and so only revealed when HK2 dissociates. This is what we address in the present paper. Furthermore, our experimental design using isolated mitochondria allows us to make important observations which would extremely difficult, if not impossible, to perform using in intact cells / in vivo:

1) we discarded cytoplasmic and mitochondrial-unbound (extramitochondrial) factors during isolation of the organelle, suggesting that another event has to occur upon HK2 release to enhance pore opening; 

2) we isolate mitochondria either from baseline or after a damaging insult ± protective regimen, ie. if there were intramitochondrial differences that could account for increased sensitivity to pore opening upon mt-HK2 release these would still be present in our setup.

In this context, we have previously shown (PMID: 21410437, 27907091) that mitochondria isolated at end-ischaemia ± IPC show similar Ca2+ accumulation while ischaemic mitochondria show increased ROS production, which are well established mPTP triggers. The fact that complete removal of mt-HK2 does not change the sensitivity of the mPTP either in the baseline or the other tested groups, which have mitochondria with modified behaviour, suggests that mt-HK2 was not acting as “the pin of a grenade”.

R#1_3. In addition, although there could be potential problems in the interventions used in previously studies as the authors claim, it is not proven that the intervention used in this study (G6P and low pH) is selective or more selective than other interventions. It would be interesting to examine whether the sensitivity of the mPTP to Ca is different in cardiomyocytes and other cells which expresses less or no HK2 (eg brain and hepatocytes) and if adding HK2 protein to isolated mitochondria change it.

A1_3 - In relation to the first point, we respect the reviewer’s position, which is similar to that of Reviewer#3 addressed below. However, we would argue that the conditions we are using to dissociate HK2, i.e. low pH and high G-6-P, are exactly those that occur in ischemia and which we have shown previously are responsible for HK2 dissociation under those conditions. We would argue strongly that this is the most appropriate conditions to use in vitro since other interventions requiring the use of synthetic peptides or drugs are not (patho)physiological and are known to have potential off target effects. Thus, we would regard such pharmacological interventions as inferior to the protocol we use. The key point in relation to our studies is that the protocol we use is highly effective at releasing HK2 and it mimics the mechanism responsible for HK2 release in ischemia. To perform all our experiments again with a potentially non-specific chemical agent would seem to us to add little for huge effort and with the potential of confusing interpretation of the data because of off target effects. We address this further under point A1_5 below.

In relation to the second point raised by the Reviewer, we agree that these are interesting ideas, but to explore them would be outside the scope of the present study. However, using a protein abundance database (PaxDB), we were able to assess the relative quantity of HK2 in different tissues of mice (such data is not available for rats). HK2 is more abundant in heart>muscle>kidney>brain>liver. To our knowledge, there is no extensive work reporting the sensitivity to mPTP in all of the tissues mentioned above. However, it is commonly accepted among mitochondriologists that mPTP sensitivity sorts as heart< brain<<liver. Although we are aware of reports showing binding of recombinant hexokinase to mitochondria (PMID: 10806396) none have evaluated mPTP sensitivity.

R#1_4. Related to point 1, what is shown here is that the dissociation does not affect Ca-induced mPTP opening, which indicates that the decrease in mtHK2 does not facilitate Ca-induced mPTP opening. This observation does not however directly address the question whether the mPTP inhibition mediated by the increase in mtHK2 is direct effect or not, a question that the authors aim to address. In addition to loss-of-function approach, gain-of-function and add-back experiments would be required to address this question.

A1_4 - The Reviewer raises two points here. With regards to the first, we would argue that by demonstrating that removal of HK2 from preconditioned ischemic mitochondria does not increase mPTP activity to the level observed in the control ischemic mitochondria we are providing evidence that the IP-mediated inhibition of mPTP opening is not caused directly by the bound HK2. In respect of the second point, it would have been interesting to add back recombinant human HK2 to isolated mitochondria, as the reviewer suggests, but we did not have the protein available and to produce it in sufficient quantity would have been a major undertaking. Furthermore, we would not have anticipated any effect of the added HK2 since removal of the HK2 already bound was without effect. Thus, we believe our conclusions remain valid without such additional experiments.

R#1_5. As I pointed out before, the conclusion achieved by the authors solely relies on a single intervention (G6P and low pH) in isolated mitochondria. The specificity of the intervention is not demonstrated as pointed above. Low pH has been suggested to affect the sensitivity of the mPTP and G6P, a product of HK catalytic activity, inhibits hexokinase activity. mtHK1 also negatively regulates mPTP opening and this could also be affected by G6P and low pH (even it is likely to be less extent).

A1_5 - The specificity of our approach has been discussed above at point A1_3. However, we would like to clarify some key aspects of our experimental setup because we feel that there has been some confusion among the reviewers (and potential future readers) during both revisions. We have endeavoured to clarify this in the present revision on page 18 by inclusion of a short statement and a new supplementary figure 1, that depicts our experimental design. 

First, the low pH / high G-6-P conditions we employed to treat isolated mitochondria were designed to mimic the conditions experienced by the mitochondria in situ during ischemia. Our treatment induces almost total dissociation of HK2 during isolation of the organelle, but during the assay of mPTP opening and other parameters (O2 consumption, swelling, calcium-retention, inner-morphology analysis, etc.) the reaction buffers were all standard and at physiological conditions (for traditional in vitro mitochondrial studies), including pH ~7.2 and in the absence of glucose or G-6-P. Therefore, although the effects mentioned by the reviewer are true, they are mimicking those experienced by mitochondria in situ and will not be present/experienced by the mitochondria during data acquisition and thus are unlikely to affect the outcome. In contrast, the few reports in the literature which have investigated the relationship between HK2 and mPTP sensitivity perform dissociation and data acquisition in the same vessel, which could account for some discrepancies between studies.

R#1_6. The authors previously showed that the TAT-HK2 dissociation impairs vascular function in Langendorff-perfused heart, although this is controversial (PMID; 23329797). The peptide has been widely used by many researchers in wide range of fields and, nonetheless, the previous observations by the authors obtained in Langendorff heart does not necessarily indicate that the HK2 dissociation peptide (without TAT sequence) has non-specific effects in isolated cardiac mitochondria.

A1_6 - We appreciate the Reviewer’s comments but would like to stress again that we chose to employ the G-6-P + low pH protocol because it mimics the physiological situation experienced by the mitochondria during ischemia and thus does not introduced potential artefacts that might be induced by synthetic peptides or drugs. We do believe that such artefacts are potentially significant. As the reviewer rightly notes the HK2_N15 peptide (without TAT sequence) might be appropriate to use in isolated mitochondria, as its sequence does not suggest it to have a penetrating character. However, assuming it acts by competing for the HK2 binding site, it might well remain bound during mitochondrial isolation and thus exert effects in its own right. Indeed, isolated cardiac mitochondria of mice overexpressing GFP-15NG show successful dissociation of mt-HK2 and retention of the recombinant protein (Fig. 4a on PMID: 31570704). So why take the risk of using the peptide when we know that G-6P and low pH work and will not continue to exert effects after isolation of the mitochondria when G-6P is removed and pH returned to normal?

R#1_7. I agree that nothing is perfectly specific, but that’s the reason why the conclusion should be supported by multiple interventions in different preparations/systems as well as by using various approaches such as gain-of-function/add-back experiments.

A1_7 - Again, we would respectfully suggest that by using the same techniques to dissociate HK2 in isolated mitochondria as we have previously shown be responsible for the dissociation in situ during ischemia, we are employing a specific and (patho)physiological relevant protocol that is not subject to any additional effects of synthetic peptides or drugs.

R#1_8. With regards to Drp1, Drp1 translocation from cytosol to mitochondria occurs through by post-transcriptional modifications including phosphorylation thus isolated mitochondria from ischemic heart (without cytosolic Drp1) is relevant to ischemic injury but not to reperfusion induced mitochondrial fission and injury. Reperfusion activates many kinases and it is shown that reperfusion increases Drp1 at mitochondria (eg., PMID; 25332205, 24477044)

The functional and mechanistic link between HK2 and mitochondrial fission proteins including Drp1 is suggested but not demonstrated by this study, although an inverse correlation is shown. As I suggested before, it used to be believed that mitochondrial fission is deleterious and fusion is protective, but recent molecular evidence suggests that this could be incorrect (PMID 25332205 21245373, 22037195). It is critical to evaluate how the changes observed in this study functionally impact on mitochondrial function/mPTP.

A1_8 - We totally agree with the Reviewer that the role and regulation of fusion / fission proteins in mPTP opening is complex and controversial and to fully investigate this interaction would be a massive undertaking that is far beyond the scope of our present study. We provide data that are suggestive of an involvement of Drp1 and Dnm2 which we hope may be a stimulus for further studies by others better equipped to perform them. We do not claim to have provided a full explanation of how everything fits together; nor do we think it is reasonable to be asked to do so. Indeed, the inconsistencies and controversies in the literature cited by us and the Reviewer suggest that this is likely to be far from straight forward! We do however discuss in some detail the complexities around the role and regulation of fusion / fission proteins on pages 31-32 and the limitations of our study in this regard on page 33. We hope the Reviewer will accept this is sufficient.

Reviewer#2 (No response)

A2_1 - We assume the Reviewer is happy with our revisions.

Reviewer#3

R#3_1. The authors have not addressed my concerns adequately. Their argument not to use a HK peptide or clotrimazole is not convincing.

A3_1 - As noted above, we would like to stress that we chose to employ the G-6-P low pH protocol because it mimics the physiological situation experienced by the mitochondria during ischemia and thus does not introduced potential artefacts that might be induced by synthetic peptides or drugs. We do believe that such artefacts are potentially significant and unknown. We have addressed in detail the reasons why we have chosen not to use synthetic peptides under points A1_3, 5 and 6 in our response to Reviewer#1. In the case of clotrimazole, to our knowledge all published papers showing that the drug releases HK2 from mitochondria have involved treating cells with clotrimazole rather than mitochondria and there is no evidence that it acts directly to displace HK2. Furthermore, in a drug-induced liver injury screening (PMID: 22987451) it was shown that clotrimazole significantly affects membrane potential and mitochondrial respiration (O2 consumption) in isolated liver mitochondria. An excerpt of Table 1 in PMID: 22987451 is shown below for the Reviewer inspection:

 Swelling ΔΨm loss Cyto c O2 cons CII O2 cons CI 

Compound Therap. class Route of admin. EC20 µM EC20 µM EC20 µM EC20 µM EC20 µM C max µM

Clotrimazole Antifungal Vaginal ND 23.9 > 800 2.9 ND 1.02

Therefore, the utility of clotrimazole for displacing mt-HK2 will depend on the effective concentration to achieve that goal versus its off-target effects. We agree we could have explored this possibility, but since we had a (patho)physiological mimic that successfully achieved our goal, we do not believe that such studies are necessary.

Reviewer#5

R#5_1. Issues adequately addressed. The study has significant limitations. But these have been appropriately acknowledged.

A5_1 - We thank the Reviewer for his approval of the revised manuscript.

---

## [Decision Letter · Decision Letter 2]

18 May 2020

PONE-D-19-26403R2

HEXOKINASE II DISSOCIATION ALONE CANNOT ACCOUNT FOR CHANGES IN HEART MITOCHONDRIAL FUNCTION, MORPHOLOGY AND SENSITIVITY TO PERMEABILITY TRANSITION PORE OPENING FOLLOWING ISCHEMIA

PLOS ONE

Dear Dr. Pereira,

Thank you for submitting your manuscript to PLOS ONE. After careful consideration, we feel that it has merit but does not fully meet PLOS ONE’s publication criteria as it currently stands. Therefore, we invite you to submit a revised version of the manuscript that addresses the points raised during the review process.

The thoughtful revision and response to Reviewer 1 is appreciated. I would like the authors to consider their responses to Reviewer 1 and decide if additional portions of these responses should be incorporated into the Introduction or Discussion. I believe that the major points in the response to reviewers from the last round have been incorporated into the current revision.

We would appreciate receiving your revised manuscript by Jul 02 2020 11:59PM. To enhance the reproducibility of your results, we recommend that if applicable you deposit your laboratory protocols in protocols.io, where a protocol can be assigned its own identifier (DOI) such that it can be cited independently in the future. For instructions see: http://journals.plos.org/plosone/s/submission-guidelines#loc-laboratory-protocols

We look forward to receiving your revised manuscript.

Kind regards,

Edward J. Lesnefsky, MD

Academic Editor

PLOS ONE

Additional Editor Comments (if provided):

The thoughtful revision and response to Reviewer 1 is appreciated. I would like the authors to consider their responses to Reviewer 1 and decide if additional portions of these responses should be incorporated into the Introduction or Discussion. I believe that the major points in the response to reviewers from the last round have been incorporated into the current revision.

Reviewers' comments:

Reviewer's Responses to Questions

**Comments to the Author**

1. If the authors have adequately addressed your comments raised in a previous round of review and you feel that this manuscript is now acceptable for publication, you may indicate that here to bypass the “Comments to the Author” section, enter your conflict of interest statement in the “Confidential to Editor” section, and submit your "Accept" recommendation.

Reviewer #1: (No Response)

Reviewer #2: All comments have been addressed

Reviewer #3: All comments have been addressed

2. Is the manuscript technically sound, and do the data support the conclusions?

Reviewer #1: No

Reviewer #2: Yes

Reviewer #3: Yes

3. Has the statistical analysis been performed appropriately and rigorously? 

Reviewer #1: Yes

Reviewer #2: Yes

Reviewer #3: Yes

4. Have the authors made all data underlying the findings in their manuscript fully available?

Reviewer #1: No

Reviewer #2: Yes

Reviewer #3: Yes

5. Is the manuscript presented in an intelligible fashion and written in standard English?

Reviewer #1: Yes

Reviewer #2: Yes

Reviewer #3: Yes

6. Review Comments to the Author

Reviewer #1: The authors did not adequately or experimentally address any of the comments that I previously raised. I still believe that there are major flaws in this work significantly affecting its relevance and final conclusions.

Reviewer #2: (No Response)

Reviewer #3: My concerns are addressed, and I do not have any additional concerns.

7. PLOS authors have the option to publish the peer review history of their article (what does this mean?). If published, this will include your full peer review and any attached files.

Reviewer #1: No

Reviewer #2: No

Reviewer #3: No

---

## [Author Response · Author response to Decision Letter 2]

28 May 2020

Thank you for your positive response to our revised paper. 

As requested, in this new revision, we have incorporated some of the points made in response to Reviewer#1 in the Introduction and Discussion.

Alterations to the original manuscript are highlighted, as per your request, in a separate file.

We hope that the manuscript is now suitable for publication in the PlosOne journal and look forward to hearing your decision.

---

## [Editor Report · Decision Letter 3]

1 Jun 2020

HEXOKINASE II DISSOCIATION ALONE CANNOT ACCOUNT FOR CHANGES IN HEART MITOCHONDRIAL FUNCTION, MORPHOLOGY AND SENSITIVITY TO PERMEABILITY TRANSITION PORE OPENING FOLLOWING ISCHEMIA

PONE-D-19-26403R3

Dear Dr. Pereira,

We are pleased to inform you that your manuscript has been judged scientifically suitable for publication and will be formally accepted for publication once it complies with all outstanding technical requirements.

With kind regards,

Edward J. Lesnefsky, MD

Academic Editor

PLOS ONE

Additional Editor Comments:The further revision of the introduction is helpful and appreciated. I believe that it will further place this work into context and increase the impact. Thank you.
---

## [Editor Report · Acceptance letter]

3 Jun 2020

PONE-D-19-26403R3 

Hexokinase II dissociation alone cannot account for changes in heart mitochondrial function, morphology and sensitivity to permeability transition pore opening following ischemia 

Dear Dr. Pereira:

I'm pleased to inform you that your manuscript has been deemed suitable for publication in PLOS ONE. Congratulations! Your manuscript is now with our production department. 

Kind regards, 

on behalf of

Dr. Edward J. Lesnefsky 

Academic Editor

PLOS ONE